



# Wind turbines in atmospheric flow - FSI simulations with hybrid LES-IDDES turbulence modelling

Christian Grinderslev[1], Niels Nørmark Sørensen[1], Sergio González Horcas[1], Niels Troldborg[1], and Frederik Zahle[1]

[1]Department of Wind Energy, Technical University of Denmark, Risø Campus, 4000, Roskilde ,

**Correspondence:** Christian Grinderslev (cgrinde@dtu.dk)

**Abstract.** In order to design future large wind turbines, knowledge is needed about the impact of aero-elasticity on the rotor loads and performance, and about the physics of the atmospheric flow surrounding the turbines. The objective of the present work is to study both effects by means of high fidelity rotor-resolved numerical simulations. In particular, unsteady computational fluid dynamics (CFD) simulations of a 2.3MW wind turbine are conducted, this rotor being the largest design with relevant experimental data available to the authors. Turbulence is modeled with two different approaches. On one hand, the well established improved delayed detached eddy simulation (IDDES) model is employed. An additional set of simulations relies on a novel hybrid turbulence model, developed within the framework of the present work. It consists on the blending of a large eddy simulation (LES) model for atmospheric flow by Deardorff with an IDDES model for the separated flow near the rotor geometry.

In the same way, the assessment of the influence of the blade flexibility is performed by comparing two different sets of computations. A first group accounts for a structural multi body dynamic (MBD) model of the blades. The MBD solver was coupled to the CFD solver during run time with a staggered fluid structure interaction (FSI) scheme. The second set of simulations uses the original rotor geometry, without accounting for any structural deflection. The results of the present work show no significant difference between the IDDES and the hybrid turbulence model. However, it is expected that future simulations of more complex stratification and longer domains will benefit from the developed hybrid model. In a similar manner, and due to the fact that the considered rotor was relatively stiff, the loading variation introduced by the blade flexibility was found to be negligible when compared to the influence of inflow turbulence. The simulation method validated here is considered highly relevant for future turbine designs, where the impact of blade elasticity will be significant and the detailed structure of the atmospheric inflow will be important.

## 1 Introduction

As future wind turbines will have unprecedented long and flexible blades, the necessity of understanding the effects of aero-elasticity on the rotor performance and on its structural integrity increases. Along with this, large wind turbines interact with a larger range of the atmospheric boundary layer, often exceeding the height of the atmospheric surface layer (ABL). This also needs consideration in the design phase, as the rotor blades consequently experience a large variation of flow through each



revolution, and flow cases which were not relevant to consider for past designs, might occur. This needed knowledge can be obtained through high fidelity methods, such as fluid-structure interaction (FSI) simulations, which model the coupled effects of both flow and structure. These simulations can further be used to develop and improve lower fidelity engineering models used by wind turbine designers in industry.

FSI of wind turbines in atmospheric turbulent flow, is not a widely studied topic, due to the computational costs of such

simulations, especially when geometrically resolved wind turbines are modelled. Instead, a more efficient manner often chosen is the use of actuator lines/discs (Sørensen and Shen, 2002; Sørensen et al., 2015). Here, the rotor is represented through body forces smeared in the CFD grid, reducing the need of grid refinement significantly. An example of actuator line based FSI in turbulent flow is the work by Lee et. al. (Lee et al., 2013), where, simulations of two aligned 5MW wind turbines in a turbulent domain modelled by LES were conducted. The structural response of the turbines was found through the FAST

aero-elastic code (Jonkman and Buhl, 2005) to study the fatigue loading. It was found that especially the surface roughness and the rotor shadow effect had large influence on the fatigue loading. As actuator lines merely represent the turbines through smeared forces, blade surface boundary layers and resulting generated wake turbulence is not modelled. Likewise, the resulting shedding of vortices at the tips and roots are not highly resolved and improperly modelled. The far wake response is, however, sufficiently accurate when the inflow to the turbine has a high turbulence intensity (Troldborg et al., 2015).

Looking at rotor resolved CFD/FSI, using LES is still too computationally expensive for many practical applications. Instead, compromises are needed for the turbulence models. In the works by Santo et al., FSI for wind turbines, structurally represented through finite element shells, were studied for steady ABL flows (Santo et al., 2020a,b) using unsteady Reynolds averaged Navier-Stokes (URANS), with the $k - \epsilon$ model. In (Santo et al., 2020a), the effects of wind shear, yaw-error, tilt and tower shadow were all investigated, finding for instance that the introduction of yaw lead to a decrease in blade deflection but a

large increase in yaw-moment on the hub. In (Santo et al., 2020b), wind gusts were introduced by acceleration of the flow near the rotor top position. One conclusion found was that for the used setup and turbulence model, a flow separation occurred when the velocity rapidly increased due to the gust, limiting the load increase avoiding any extreme deflections. To consider turbulent fluctuating flow, a popular alternative to LES are the synthetic turbulence generators such as the method by Mann (Mann, 1998). These methods efficiently create boxes of turbulent fluctuations, that can be used to create inflow for CFD

simulations or inserted internally in the domain by additional body forces (Troldborg et al., 2014). Along with this, a hybrid turbulence model like detached eddy simulation (DES) can be used to resolve the turbulence in the grid. This model combines the URANS approach for attached flow regions with LES in the separated regions. The use of synthetic turbulence is efficient as the modelling of turbulent fluctuations is fast, and the DES models need less grid resolution near the rotor than LES. The turbulence will, however, not be in balance with the CFD simulation shear as shown in (Troldborg et al., 2014), and thereby the

turbulence will change as it convects through the domain. Another drawback of this method is that the modelled turbulence is neutrally stratified and therefore it cannot naturally handle atmospheric stability. Further, a potential problem of the synthetic turbulence methods is the assumption of homogeneous and Gaussian turbulence. Even though previous work (Berg et al., 2016) have shown that the latter assumption does not significantly affect the loads on a wind turbine under normal conditions one could easily come across cases where these assumptions does not hold. In (Li et al., 2015) synthetic turbulence was used



to study the geometrically resolved NREL 5MW reference turbine in sheared and turbulent inflow including flexibility of
the rotor. The FSI framework was based on a CFD solver coupled with a multi-body dynamics (MBD) structural solver and
turbulence was imposed at the inlet using the Mann turbulence box as input. The main conclusions of the study was that
realistic atmospheric flow including shear and turbulence is important when designing large scale wind turbines in terms of
loading. Additionally, the study concluded that, for the specific turbine and flow cases, inclusion of blade flexibility does not

impact highly the wake behaviour, whereas inflow turbulence have high impacts on wake diffusion. Guma et al. (Guma et al.,
2020) recently published a study looking into the aero-elastic response of the NM80 rotor, also studied in the present article,
in turbulent inflow. Here, synthetic Mann box turbulence and the delayed detached eddy simulation (DDES) turbulence model
were used to create and resolve turbulent structures in the wind flow. The fluctuating forces occurring on the blades were used
to calculate the fatigue damage on the blades by means of the so-called damage equivalent loading (DEL). It is found that for

low inflow velocities, the DEL is mainly influenced by the turbulent inflow rather than the inclusion of flexibility, at least for
the considered relatively stiff rotor.

An alternative more complex method to simulate geometrically resolved turbines in the ABL flow was proposed by Vi-
jayakumar et. al. (Vijayakumar, 2015). Here, a hybrid turbulence model was developed, which combines spectral ABL LES
simulations by Moeng (Moeng, 1984) with more feasible URANS based $k - \omega$ SAS (Egorov et al., 2010) simulations close

to the rotor. By this, a large decrease of grid cells is needed (however still large), as the URANS based turbulence models the
effect of all the turbulent scales. The model was studied on a single wind turbine blade in (Vijayakumar et al.), however, using
pure CFD without a structural coupling.

In general, considering presently available high-performance computing capabilities, compromises are needed when doing
high fidelity aero-elastic modelling of wind turbines in atmospheric flow using FSI. This being either by reducing the rotor

representation by actuator lines to allow LES simulations, or instead simplifying the turbulence modelling.

The objective of the present study is to move one step up the ladder of complexity by investigating rotor aerodynamics and
aero-elasticity in turbulent LES inflow, using a novel turbulence model. The model is inspired by the one of Vijayakumar,
combining the ABL turbulent flow modelling of the Deardorff LES model with the IDDES engineering model near the rotor.
The study will be done through blade resolved FSI simulations of the 2.3 MW NM80 wind turbine rotor including blade

flexibility using a FSI coupling framework combining the computational fluid dynamics (CFD) code EllipSys3D (Michelsen,
1992, 1994; Sørensen, 1995) and the structural solver from the aero-elastic code HAWC2 (Larsen and Hansen, 2007). For the
specific rotor, measurements of inflow and blade loading are available for validation of results. The study is a continuation of
(Grinderslev et al.), where FSI of the NM80 rotor was studied in sheared and yawed, however laminar, inflow.

## 2    Methodology

In this section, the computational solvers are presented along with the simulation strategies such as FSI framework and pre-
cursor simulations. Further, the hybrid turbulence model will be introduced by first introducing the participating turbulence
models. Finally, the computational grids used in the study are described along with the chosen simulation parameters.





### 2.1 Numerical methods

#### 2.1.1 Flow solver

To solve the fluid flow, the DTU inhouse CFD code EllipSys3D (Michelsen, 1992, 1994; Sørensen, 1995) is used. The code solves the incompressible Navier-Stokes equations in structured curvilinear coordinates using the finite volume method with a collocated grid arrangement. The code is parallel and highly scalable using the message passing interface (MPI) and multi-block decomposition, the multi-grid method and grid sequencing. EllipSys3D has multiple convective schemes implemented, such as central difference (CDS), second order upwind (SUDS) and quadratic upstream interpolation for convective kinemat-

ics (QUICK). For solution of the pressure correction equation, various algorithms are implemented such as PISO, SIMPLE, SIMPLEC and variations hereof. Rhie-Chow interpolation is used to avoid odd/even pressure decoupling. Overset capabilities, including grid hole-cutting are implemented internally in the code (Zahle et al., 2009).

Several turbulence models are implemented such as two equation Reynolds Averaged Navier Stokes (RANS) models, $k - \epsilon$ and $k - \omega$ among others, hybrid models like detached eddy simulations (DES), delayed DES (DDES), improved DDES (IDDES)

and multiple large eddy simulations (LES) models. In addition to these, a hybrid version of the LES and the IDDES model will be presented in this paper.

For FSI simulations, the deformation of grids is handled through a moving mesh method with a volume blend factor, which propagates the surface displacement along grid lines normal to the surface blended to the original volume proportionally to the distance to the blade surface. This ensures that mesh points in the vicinity of the blade surface are displaced as a solid body

movement along with the blade, while points further away only move a fraction of the displacement, using a blending function. When using the overset grid method, the deformation is only transferred to the volume grid blocks containing the solid surface.

The code has been used extensively for for a range of test cases and was validated in e.g the Mexico project (Bechmann et al., 2011; Sørensen et al., 2016) and for the Phase VI NREL rotor (Sørensen and Schreck, 2014; Sørensen et al., 2002). Recently, the code was validated in (Grinderslev et al., 2020b) for the specific case of the present NM80 rotor in atmospheric laminar

flow conditions by comparison with the CFD code Nalu-Wind (Sprague et al., 2019) and measurements from the DanAero experiments. Further, the FSI framework was used in (Grinderslev et al.) to simulate the coupled effects of the DanAero inspired laminar wind flow and the structural response.

#### 2.1.2 Aero-elastic solver

HAWC2 (Larsen and Hansen, 2007) is an aero-elastic code used for calculating blade element momentum (BEM) aerody-

namics and structural responses of wind turbines. The structural part of the code is based on the multi-body dynamics (MBD) formulation, accounting for non-linear effects of large deflections. Each structural component, i.e. a blade or the tower, can be represented by a number of bodies assembled by linear Euler or Timochenko beam elements. Sub-bodies are connected with constraint equations considering non-linearities.

HAWC2 has a built-in aerodynamics module, that calculates aerodynamic forces using BEM theory. As is common in BEM

implementations, prediction of airfoil aerodynamic performance is based on pre-computed look-up tables of lift, drag and



moment, which is needed to calculate forces along the blade. Multiple correction schemes are implemented to improve the BEM aerodynamics such as tip loss corrections, dynamic stall models, tower shadow effect and much more, see (Madsen et al., 2019).

HAWC2 is widely used by industry, and has been verified and validated in various studies (Pavese et al., 2015; Madsen et al., 2019), considering the structural and aerodynamics aspects of the code respectively.

### 2.1.3 FSI-framework

The two codes EllipSys3D and HAWC2 are coupled, in a partitioned manner, through the Python framework, referred to as the DTU coupling, originally created by Heinz (Heinz et al., 2016a) and further developed by G. Horcas and R. Garcia (Horcas et al., 2019; García Ramos et al., 2020). Through the use of the coupling framework, the BEM aerodynamics module of HAWC2 is replaced by an interface to the EllipSys3D CFD code.

Using predicted displacements of nodes from HAWC2, the CFD mesh is deformed, and a new flow-field is found through EllipSys3D. The loads predicted by the CFD solver are then applied to the HAWC2 structural model and a new deformation is found. All communication between EllipSys3D and HAWC2 happens through the DTU coupling framework. In (Heinz et al., 2016a), a loosely coupled approach was found to be sufficient for wind energy related cases, due to the high mass ratio between the turbine structure and air, and is therefore used.

Studies involving the application of the FSI framework, for both operational and standstill configurations, include (Heinz et al., 2016a,b; Horcas et al., 2019, 2020). The framework has been validated with experiments through simulations of a pull-release test of a wind turbine blade in the large scale test facility of DTU, see (Grinderslev et al., 2020a). The process of the framework between the main iterations can be described through the following steps:

– The displacements of the present time step are predicted by HAWC2 with second order accuracy, using kinematics from the previous time step.

– Displacements are sent to EllipSys3D and the surface mesh is deformed while displacements are propagated into the volume mesh using a volume blend method.

– The Navier-Stokes equations are solved to calculate the flow field for the new time step through under-relaxed sub-iterations in EllipSys3D.

– Forces are computed and integrated on the CFD mesh surface and sent to HAWC2.

– Forces are interpolated to the aerodynamic sections of the HAWC2 model and actual deformations are calculated.

– Unless the solution has reached the total simulation time, the simulation is advanced to the next time step and the procedure is repeated.





## 2.2 Turbulence modelling

A hybrid turbulence model has been developed to consider the dominant turbulence scales from the atmospheric boundary layer (ABL) down to the blade boundary layer (BBL), within the same simulation. To do this, the Deardorff one-equation LES turbulence model for ABL flows (Deardorff, 1972) is blended with the IDDES turbulence model (Shur et al., 2008), which itself is a blend between URANS modelling in the BBL and LES in the separation region outside the BBL. The blending of the two models happens through the energy equation, which is solved for in both methods. In the Deardorff model, the transport equation of sub grid scale (SGS) energy $e$ is solved, whereas the transport equation for total turbulent kinetic energy $k$ is solved in the IDDES method. These energy expressions are blended through their respective terms of diffusion, convection, production, buoyancy and dissipation using a smooth tanh blending function. By this, $e$ of the Deardorff model coming towards the rotor is transformed into equivalent $k$ of the IDDES, and vice versa in the wake region. In the following, the two models will be introduced, followed by a description of the blending for the hybrid model used in this study.

### 2.2.1 Deardorff large eddy simulation model

In the Deardorff LES turbulence model (Deardorff, 1972, 1980), the turbulent eddy viscosity $\mu_t$ is calculated through the expression:

$$\mu_t = C_k \rho l_{LES} \sqrt{e} \tag{1}$$

Here, $C_k$ is a constant of 0.1, $\rho$ is the air density and $l_{LES}$ is a mixing length scale, which for neutral stratification is set equal $\Delta_{LES}$ , being the grid size, here defined as $\Delta_{LES} = (dx \cdot dy \cdot dz)^{1/3}$, $dx$, $dy$ and $dz$ being the grid spacing in the respective directions.

The SGS energy, $e$, is found by solving the following transport equation :

$$\frac{D\rho e}{Dt} = -\tau_{ij}S_{ij} + \frac{g}{\theta_0}\tau_{\theta w,LES} - C_\epsilon \rho \frac{e^{3/2}}{l_{LES}} + \frac{\partial}{\partial x_j}\left((\mu + 2\mu_t)\frac{\partial e}{\partial x_j}\right) \tag{2}$$

where $g$, $t$ and $\mu$ refer to the gravity, time and molecular viscosity respectively. $C_\epsilon$ is equal to 0.93, the buoyancy SGS fluxes $\tau_{\theta i,LES} = -\mu_\theta \frac{\partial \theta}{\partial x_i}$, with variable temperature $\theta$ and the eddy heat diffusivity being: $\mu_\theta = \left(1 + \frac{2l_{LES}}{\Delta_{LES}}\right)\mu_t$. $\theta_0$ is the surface reference temperature. The SGS stress tensor $\tau_{ij}$ is defined as; $\tau_{ij} = -2\mu_t S_{ij}$ using the strain rate $S_{ij} = 1/2\left(\frac{\partial u_i}{\partial x_j} + \frac{\partial u_j}{\partial x_i}\right)$, with $u$ being the velocity vector.

### 2.2.2 SST based detached eddy simulation models

For the $k-\omega$ SST based DES turbulence models, $\mu_t$ is found through the standard $k-\omega$ SST (Menter, 1993) approach, which is then altered in the dissipation term depending on the chosen DES model.

$$\mu_t = \rho \frac{a_1 k}{\max(a_1\omega, \Omega F_2)}, \quad \text{with} \quad a_1 = 0.31 \tag{3}$$

Here, $k$ is a total turbulent kinetic energy, $\omega$ the specific dissipation rate, $\Omega$ the shear-strain rate, and $F_2$ a limiting blending function. $k$ and $\omega$ are found through the following two transport equations:





For $k$:

$$\frac{D\rho k}{Dt} = -\tau_{ij}S_{ij} + \frac{g}{\theta_0}\tau_{\theta w,DES} - \rho\frac{k^{3/2}}{\tilde{l}} + \frac{\partial}{\partial x_j}\left[(\mu + \mu_t\sigma_k)\frac{\partial k}{\partial x_j}\right] \tag{4}$$

For $\omega$:

$$\frac{D\rho\omega}{Dt} = \frac{\gamma}{\nu_t}\tau_{ij}\frac{\partial u_i}{\partial x_j} - \rho\beta\omega^2 + \frac{\partial}{\partial x_j}\left[(\mu + \mu_t\sigma_\omega)\frac{\partial\omega}{\partial x_j}\right] + 2(1-F_1)\rho\sigma_{\omega 2}\frac{1}{\omega}\frac{\partial k}{\partial x_i}\frac{\partial\omega}{\partial x_i} \tag{5}$$

Here $F_1$ is a blending function, to shift between the standard $k-\omega$ model ($F_1 = 1$) near the surface to the $k-\epsilon$ model ($F_1 = 0$) within the boundary layer and further out, while $\sigma_k$, $\sigma_\omega$, $\beta$ and $\gamma$ are parameters which themselves depend on $F_1$. Finally $\beta^*$ and $\sigma_{\omega 2}$ are constants. $\nu_t$ is the kinematic turbulent viscosity $\nu_t = \mu_t/\rho$. All constants and parameters as well as blending functions $F_1$ and $F_2$ can be found in the original work by Menter (Menter, 1993). In the original work by Menter, the buoyancy term is not considered. In the EllipSys3D version of the $k-\omega$ SST, it is however considered as the second term of the right hand side of Eq. (4), and the flux is found as: $\tau_{\theta i,DES} = -\mu_t\frac{\partial\theta}{\partial x_i}$

The length scale which appears in the $k$-equation serves to switch from URANS to LES mode and is defined as:

$$\tilde{l} = \min(l_{k-\omega}, l_{DES}), \tag{6}$$

where $l_{k-\omega} = \sqrt{k}/(\beta^*\omega)$ and $l_{DES}$ is the length scale in the LES region. In the standard DES model (Spalart et al., 1997; Travin et al., 2004), $l_{DES} = C_{DES}\Delta_{DES}$, where $\Delta_{DES} = \max(dx, dy, dz)$ and $C_{DES}$ is a $F_1$ dependent parameter.

DES is known to be sensitive to sudden changes of grid refinements as grid induced separation (GIS) can be introduced. Here, the modelled turbulent viscosity will drop instantly without the additional turbulence being resolved. It is also known to have a mismatch between the URANS and LES region, if used as a wall modelled LES model. These issues are addressed in delayed DES (DDES) (Menter et al., 2003; Spalart et al., 2006), improved DDES (IDDES) (Shur et al., 2008) and simplified improved DES (SIDDES) (Gritskevich et al., 2012), by using more advanced expressions for the length scale $\tilde{l}$.

### 2.2.3 Hybrid ABL - BBL model

In order to simulate the effect of turbulence in both ABL and BBL scales, a hybrid method is suggested where the Deardorff ABL LES model is blended together with the BBL DES models to avoid the need of excessive grid resolution in the BBL otherwise needed by LES. The blending is established through a blending function $F_h$ which is zero in the ABL region and one in the DES region and then defining a hybrid turbulence kinetic energy $\tilde{k} = F_h k + (1 - F_h)e$. Using these definitions the energy equations Eq. (2) and (4) are combined to give the following transport equation for $\tilde{k}$

$$\frac{D\rho\tilde{k}}{Dt} = -\tau_{ij}S_{ij} + \frac{g}{\theta_0}(F_h\tau_{\theta w,DES} + (1-F_h)\tau_{\theta w,LES}) - \rho\tilde{k}^{3/2}\left(\frac{F_h}{\tilde{l}} - C_\epsilon\frac{1-F_h}{l_{LES}}\right)$$

$$+ \frac{\partial}{\partial x_j}\left[(\mu + \mu_t(\sigma_k F_h + 2(1-F_h)))\frac{\partial\tilde{k}}{\partial x_j}\right] \tag{7}$$

The blending function $F_h$ is defined as follows:

$$F_h = 0.5 - 0.5\cdot\left(\tanh\left((d_w - R)\cdot\frac{2}{\delta_{\text{blend}}}\right)\right) \tag{8}$$





Here, $d_w$ is the wall normal distance, $R$ is the wall distance to the location where $F_h$=0.5 and $\delta_{blend}$ is the transition distance between $F_h$=0.12 and $F_h$=0.88, where the blend is most rapid.

To allow the present method to work together with the $k-\omega$ model, an expression for $\omega$ is needed in the LES region. This expression is made through the standard $k-\omega$ turbulent viscosity expression, to ensure consistency through the blending regimes.

$$\mu_t = C_k \rho l_{LES} \sqrt{\tilde{k}} = \rho \frac{\tilde{k}}{\omega_{LES}} \Rightarrow \omega_{LES} = \frac{\sqrt{\tilde{k}}}{C_k l_{LES}} \tag{9}$$

Here, it is assumed that the blending from DES to LES happens in the region where $F_2 = 0$, such that the viscosity limiter is

inactive. A blended expression $\tilde{\omega}$ is then found for the entire domain.

$$\tilde{\omega} = \omega F_h + \omega_{LES}(1 - F_h) \tag{10}$$

This allows the calculation of the turbulent viscosity similar to Eq. (3):

$$\mu_t = \rho \frac{a_1 \tilde{k}}{\max(a_1 \tilde{\omega}, F_2 \Omega)} \tag{11}$$

It is noted, that in the Deardorff part of the model, the turbulent viscosity, $\mu_t$, is linearly proportional to the length scale,

$l_{LES}$, through $\omega_{LES}$, see Eq. (9). This needs to be considered if sudden changes are made to the grid resolution, as this will lead to a proportionally equal change to $\mu_t$. This could for instance be the case with overset grids, as used in the present study, where sudden changes of grid resolutions are happening over the interface. This should in theory be fine, as the resolved turbulence adapts to the grid, but as seen with the known GIS issues of the original DES model, the change in resolved turbulence, due to grid refinement, does not happen instantaneously. In the present study, this is handled by limiting the LES length scale $\Delta_{LES}$

to the grid size of the background grid, see Section 2.3.1.

### 2.2.4   Turbulent inflow simulations

In this study, the turbulent flow of the atmospheric boundary layer is modelled through a LES precursor simulation using the Deardorff model. Here, a neutrally stratified wind profile is simulated and sampled for use as input in the successor simulation including the rotor.

In the successor simulation the hybrid LES/IDDES model is used. LES is used for turbulence modelling in the majority of the domain, except for close to the rotor. In this area, the IDDES model is utilized instead, which requires less grid resolution near the rotor surface than LES.

The precursor conditions are approximating measurements from the DanAero field experiment (Bak et al., 2010), where a met mast located ≈2.5 diameters from the considered rotor measured the wind field using cup-anemometers at five points

vertically at 17m, 28.5m, 41m, 57m (hub height), 77m and 93m. The data-set from these cup-anemometers is used to fit a corresponding neutral log-law wind profile to generate inputs for the Schumann-Grötzbach wall model (Schumann, 1975;





Grötzbach, 1987) used in the simulation.

$$U = u_* / \kappa \cdot \ln(z/z_0) \tag{12}$$

where $U$ is the wind speed, $u_*$ is the friction velocity, $\kappa$ the Von Karman constant ($\approx 0.4$), $z$ the vertical coordinate and finally
$z_0$ the roughness length. As a neutral stratification flow is modelled for simplicity, no temperature is modelled in the present
study.

### 2.3   Simulation setups

#### 2.3.1   EllipSys3D model

Air is described with density of 1.22 kg m$^{-3}$ and a dynamic viscosity of $1.769 \cdot 10^{-5}$ kg m$^{-1}$ s$^{-1}$. The convective terms
are calculated through a blend of the fourth order central difference (CDS4) scheme in the LES area and the upwind QUICK
scheme in the URANS part as described in (Strelets, 2001). An improved version of the the SIMPLEC algorithm (Shen et al.,
2003) is used to couple the velocity and pressure. No transition model is applied, such that the blade boundary layer is assumed
fully turbulent. A time increment corresponding to 0.125° rotation per time step is used for all simulations corresponding to
$1.29 \cdot 10^{-3}$ seconds per time step.

#### 2.3.2   Turbulence blending

To enable the hybrid turbulence modelling, a blending region must be defined. As mentioned in Section 2.2.3, a sudden grid
refinement will create a sudden length scale change, and thereby, if in the Deardorff LES region, a sudden change of turbulent
viscosity. In the present setup with overset grids, it is therefore chosen to avoid the viscosity "jump" by keeping the LES length
scale $\Delta_{LES}$ to the background grid value. By this, the refinement does not change the dissipation length scale nor the viscosity.
Near the rotor however, an IDDES zone is prescribed depending on the wall distance. The blending happens 8m from the
surface with the majority of the blend happening over a 4m distance to ensure a smooth transfer from LES to IDDES, see
Figure 1. In the IDDES region, the refined mesh impacts the turbulent dissipation through $\Delta_{DES}$, as usual. By this, the small
scale detached flow is still captured close to the rotor.





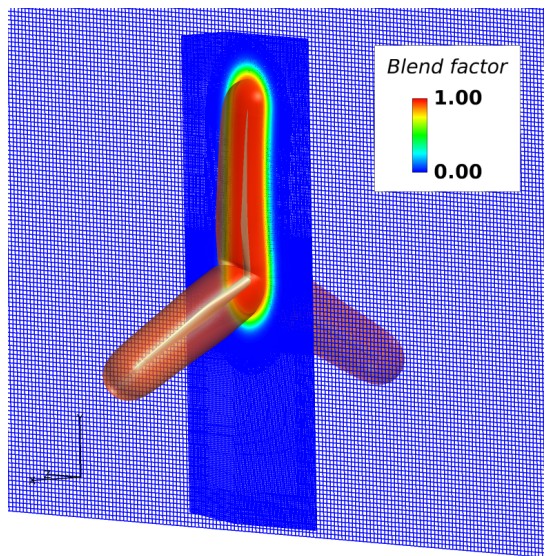

**Figure 1.** Blend factor, $F_h$, definition. Red: IDDES region, Blue: Deardorff LES region. Isosurface of blend factor $F_h$=0.9

### 265 2.3.3 HAWC2 model

The structural model used for the NM80 turbine has been created and validated internally at DTU Wind Energy as part of the original DanAero project (Bak et al., 2013). This model was also used in a former study of FSI on the same rotor (Grinderslev et al.) comparing URANS FSI with BEM based aero-elastic simulations for complex laminar flow scenarios. The blade has a prebend into the wind of ≈1.5m at the tip. Each blade is structurally discretized into 22 bodies each consisting of one

Timochenko beam element. As only the rotor is modelled in CFD, only blade flexibility is considered as well. This means that tower and shaft are considered rigid in the structural model and no tower shadow is considered. A total of 60 aerodynamic sections are distributed per blade, which are used for both BEM and CFD loads. For the initialization of the FSI simulation, BEM calculations are run in the aero-elastic code to reach the time of the initialized CFD simulation which is run alone to obtain an initial flow field before enabling the coupling. For this, airfoil data is used, obtained during the original DanAero

project through wind tunnel tests and corrected for 3D effects, see (Bak et al., 2006) and (Bak et al., 2011). From (Grinderslev et al.), it is known that the airfoil data does not capture well the 3D effects, and predicts an earlier stall than seen in CFD or experiments. In this case, however, the BEM calculations are used for initialization only to get good guesses on initial bending, and for that reason no further corrections to the airfoil data have been conducted. Dynamic stall corrections (Hansen et al., 2004), and tip corrections (Glauert, 1935) are applied during the initializing BEM calculation. No controller is used, as

a constant rotation speed of 16.2 rpm and pitch setting of -4.75° (decreasing the angle of attack) is set. For simplicity, the yaw and tilt is omitted in the simulation setup. For the DanAero campaign used for comparison, a tilt of 5° and average yaw error of 6.01° were present, however.





## 2.4 Simulation method

### 2.4.1 Precursor simulation

For the precursor simulation, as a first step, the turbulent flow is developed by recycling the flow using periodic boundary conditions. This resembles the flow moving over a very long distance, building up the boundary layer and producing the turbulence through shear production. In order to ensure a mean profile close to the desired measured wind velocity profile, the SG wall model is used. This forces the surface shear stress of the first adjacent cells to the ground to fit the log-law.

Initially, the grid sequencing scheme of EllipSys3D is utilized on three grid levels to speed up the simulation and reach 290 a fully turbulent domain quickly. When the flow is fully turbulent and the mean flow profiles match the desired flow, planes consisting of velocity components, $U$,$V$,$W$, pressure, $P$, and SGS kinetic energy $e$ are sampled. The plane is centered in the cross flow directions of 1000m×600m with 4m cell distances, see Figure 2.

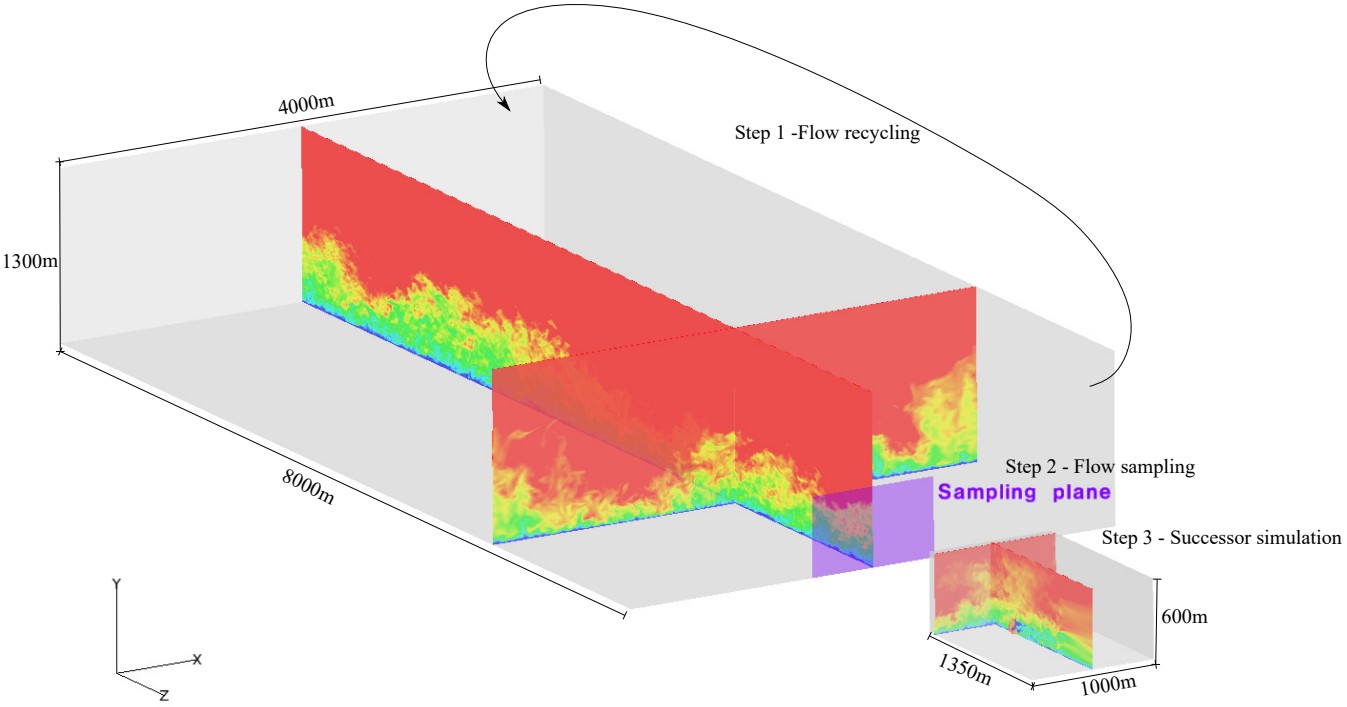

**Figure 2.** Concept of precursor to successor simulations along with domain sizes of conducted precursor and successor simulations

### 2.4.2 FSI simulation

The FSI successor simulation process is divided into phases depicted in Figure 3.

In the first phase, simulations without coupling to the structural solver are run to develop the flow and fill the domain with the sampled turbulent flow. In this phase, the grid sequencing scheme of EllipSys3D is used exploring coarser grids to minimize





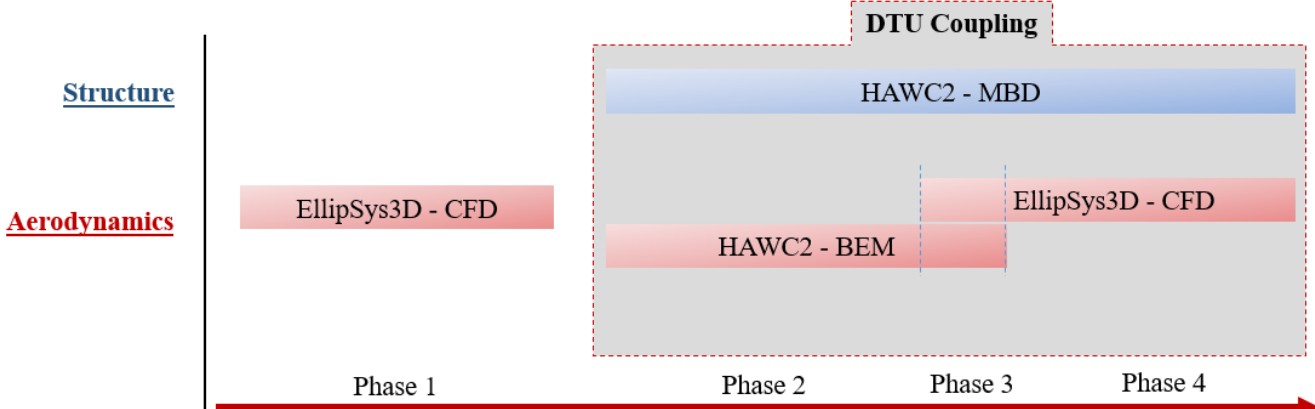

**Figure 3.** Process diagram of conducted simulations

the simulation cost during spin-up.

When passing to the FSI framework, phase 2, HAWC2 is run for the same amount of revolutions using BEM aerodynamics corresponding to the mean flow profile to ensure compatibility in time between the solvers when coupling, and obtaining a good guess of initial blade deformations. In phase 3 the coupling of EllipSys3D and HAWC2 is initiated with a smooth linear blending of forces over two revolutions to switch from BEM to CFD loading. This avoids any large force jumps in the HAWC2 solver, such that no undesired vibrations are introduced to the system. In the final phase 4, CFD loads are used for calculation of structural response and a full 2-way coupling is simulated for the desired amount of revolutions.

### 2.5 Computational grids

#### 2.5.1 Precursor simulation

The precursor domain is $4000\times1300\times8000$ meters (width$\times$height$\times$length) discretized $576\times256\times1920$ cells divided in 8640 blocks of $32^3$ cells. A total of $\approx$283M cells are present in the precursor. The grid cells vary in size in the cross flow directions to obtain higher resolution in the sampling area. In the sampling area, the cells are cubic with 4m cell sides, while cells are slowly stretched towards the boundary sides and top. Periodic boundaries are prescribed on the vertical sides while a symmetry condition is used on the top boundary and the Schumann-Grötzbach (Schumann, 1975; Grötzbach, 1987) (SG) wall model is used for ensuring the Monin-Obukhov similarity law in the first cells adjacent to the wall by dictating the wall shear stress.

#### 2.5.2 Successor simulation

For the rotor simulations, an overset grid method is utilized (Zahle et al., 2009), as this allows for a stationary background grid including the ground, while a rotating grid can be used for the rotor. Flow information is then communicated by interpolation between the grids through donor and receiver cells within the overlapping region of the meshes.





In the present setup, only the rotor is considered, omitting the tower and nacelle, with a total of three overlapping mesh groups, see Figure 4. Near the rotor, an O-O type mesh is grown from the blade surface, extruding ≈15m, discretized over 128 cells from the surface using the mesh tool HypGrid (Sørensen, 1998). The first cell adjacent to the rotor surface has a height of $1 \cdot 10^{-6}$m to ensure a $y^+$ of less than 1. Each blade is represented through 128 grid points spanwise and 256 chordwise. The
blade tip and grid around a blade section are presented in Figure 5. The rotor diameter, D, is ≈80m.

Around the rotor mesh, a cylindric disc mesh is constructed with pre-cut holes around the blades. This mesh rotates along with the rotor mesh, speeding up the hole-cutting algorithm, as the holes move along with the rotor.

To simplify the overset search of donor and receiver cells and speed up computations, all deformation from the rotor will be propagated to the rotor mesh in such a way, that only cells that are inside the hole region of the overlapping disc mesh deform.
Through this simplification, there is no need for updating communication tables for donor and receiver cells between the rotor and disc mesh, as the two rotate together. This choice necessitates the hole of the disc mesh to be far enough from the surface to leave room for the deformation of mesh cells without impairing the cell quality. The disc and rotor grids are similar to the setup used in (Grinderslev et al.), however in this study the background grid has changed to be suitable for LES simulations.

The background domain is a box of 1000 (12.5D) × 600 (7.5D) × 1350 (16.9D) meters (width × height × length) using
352×256×640 cells adding up to ≈58M cells. A concentration of cells is present in the cross flow directions around the rotor area down to 1m side lengths, see Figure 4 (right). Cells in the flow direction are kept constant of ≈1.4m from inlet to the rotor and 6D behind it, before stretching towards the outlet. Boundary conditions are velocity inlet, outlet assuming fully developed flow, and symmetry conditions (slip walls) on sides and top boundaries. The ground has a no-slip wall condition, but with the SG wall model as in the precursor simulation. The rotor is placed ≈4.38D from the inlet, ≈6.25D from sides and top and
≈12.5D from the outlet.

A total of 78M cells are used for the combined setup.

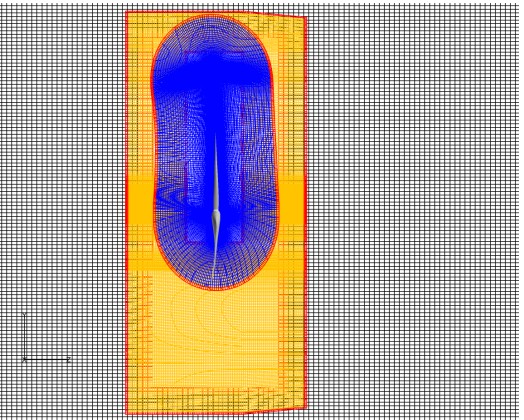
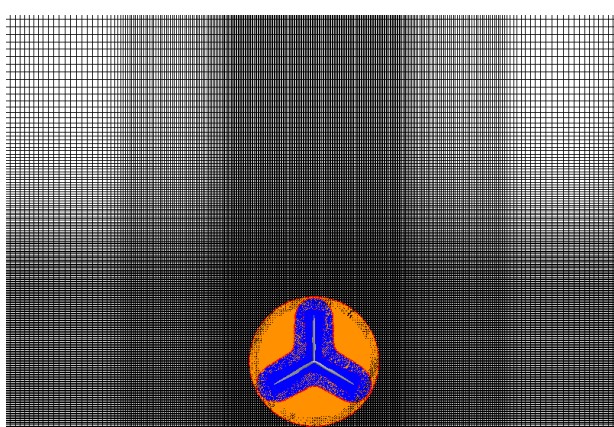

**Figure 4.** Grids used for simulations. Left: side view, right: front view. Red cells show receiver cells of overlapping grids. Blue: rotor grid, orange: disc grid, black: background grid. Entire background grid is not shown.



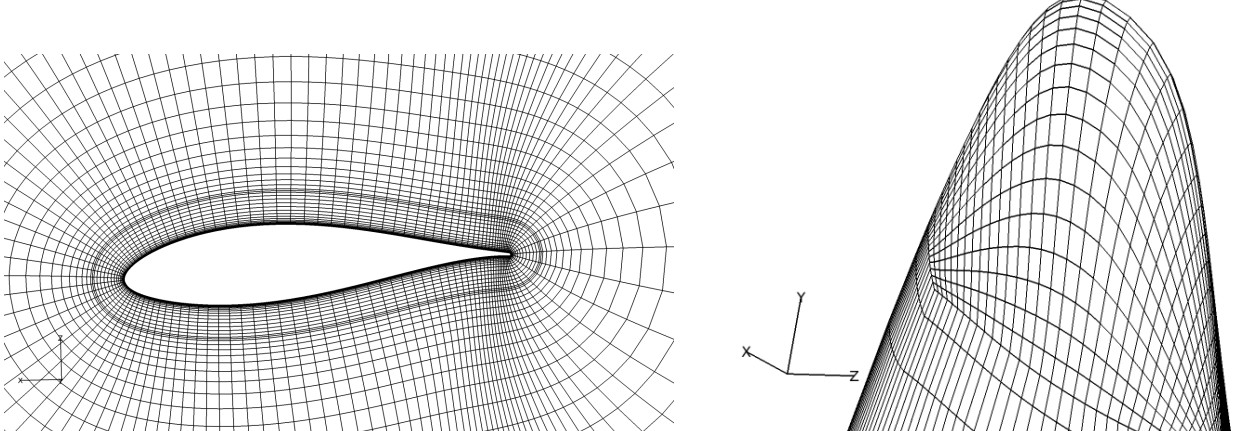

**Figure 5.** Near rotor mesh at 25m span and surface discretization at tip. Only every second line shown

## 3 Results

### 3.1 Precursor simulation

A total of 9750 seconds were simulated for the precursor simulation, of which the final 1000 seconds were sampled, in a

period where the developed flow profile sufficiently matched the desired profile. The precursor was run in three grid levels with varying time steps. First, the coarse period ($\delta z=\delta y=\delta x \approx 16m$, $\Delta t=1.0$ sec), a medium period ($\delta z=\delta y=\delta x \approx 8m$, $\Delta t=0.5$ sec) and finally a fine period ($\delta z=\delta y=\delta x \approx 4m$, $\Delta t=0.25$ sec) of which the sampling was conducted as depicted in Figure 6 along with the spectra at three different altitudes. As seen, the turbulence is well resolved with a long inertial subrange following the Kolmogorov spectrum law with a decaying slope of -5/3.

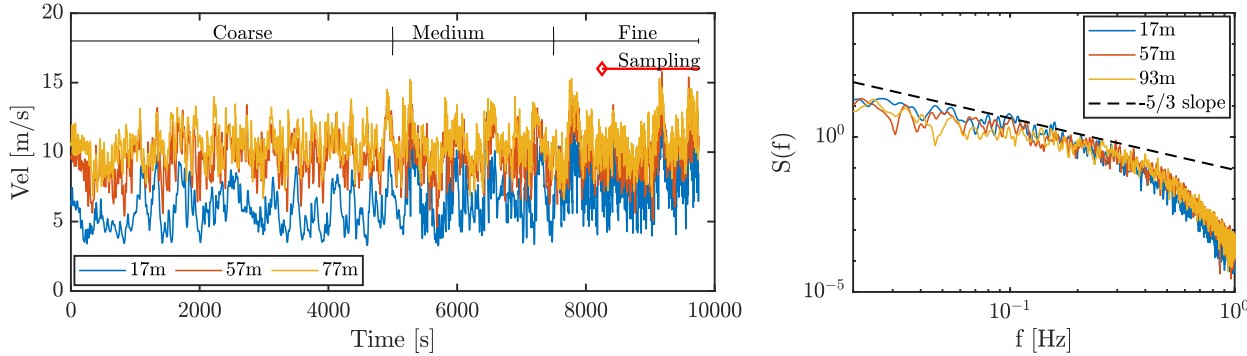

**Figure 6.** Left: Time series of wind speed, $W$, at three altitudes approximately matching the rotor bottom, center and top altitudes. Right: spectra of wind speed time series (fine resolution period only) using the Welsh estimate





From the sampling plane, the wind speed profiles of $W$ were extracted and horizontally and temporally averaged $\pm$ 1D from the rotor position in the cross plane direction as depicted in Figure 7 (left). As seen, the relative difference of the averaged profile and the DanAero log-law fit match well with a maximum of 8% at $\approx$ 14m, which corresponds to only $\approx$0.5m/s at the specific altitude. One difference to note, however, is the larger standard deviation, and thereby turbulence intensity, of the sampled flow, with fluctuations that supersede the DanAero measurements. The complexity of fitting both mean profile

and turbulence intensity between measurements and LES simulations is high. In this specific case, the assumption of neutral stratification in the simulation, while no knowledge about stratification being available from the measurements, likely plays a role in the capabilities to match results. This was the best match obtained after multiple calibration attempts, considering both mean profile and turbulence intensity.

Figure 8 depicts the resulting resolved and SGS flow shear stresses and resolved friction velocity, $u^*$.

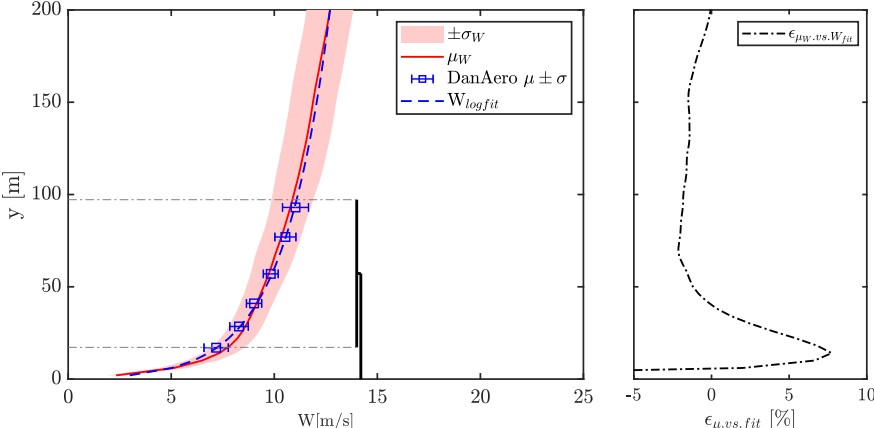

**Figure 7.** Left: Horizontal and temporal average profile $\mu_W \pm 1 \sigma$ (solid red and red patch), DanAero measurements and fitted log-law (blue errorbars and dashed). Horizontal averaging based on flow from $\pm$1D from the rotor center on the sampled flow plane. Right: Relative error between log-law fit and $\mu_W$ profile





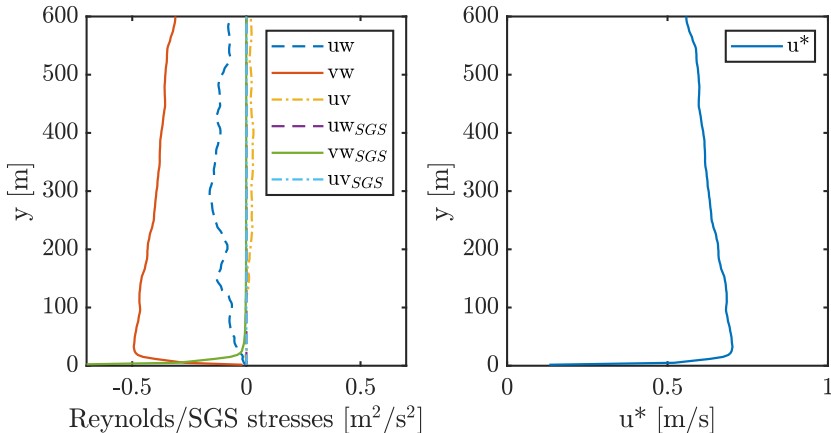

**Figure 8.** Left:Horizontal averages of resolved and SGS shear stresses at 9250 sec. Right: Resulting resolved friction velocity $(u^*)^2 = \sqrt{vw^2 + uv^2}$

### 3.2 Successor simulation

In the following, the results of the successor simulations are presented. First, the new turbulence model is compared to the same setup using only the IDDES turbulence model. Further, results from simulations using the hybrid model with and without flexibility of the blades are presented to study the effect of the blade elasticity.

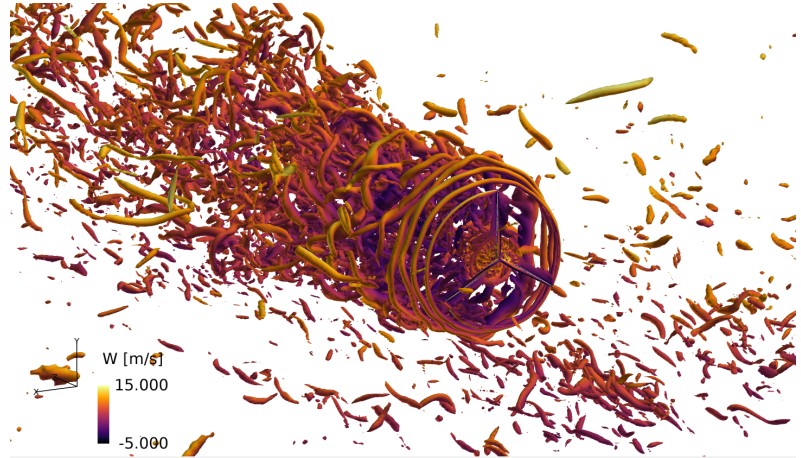

**Figure 9.** Isobars of Q-criterion=0.4 colored with value of flow velocity $W$

Figure 9, shows the Q-criterion=0.4 (Hunt et al., 1988) of the flow, visualizing the turbulent structures up- and especially downwind of the rotor. As seen, the tip vortices in the wake are quickly broken up into smaller structures by the surrounding turbulent flow. This is also visible in Figure 10 showing the velocity field at multiple downstream positions.





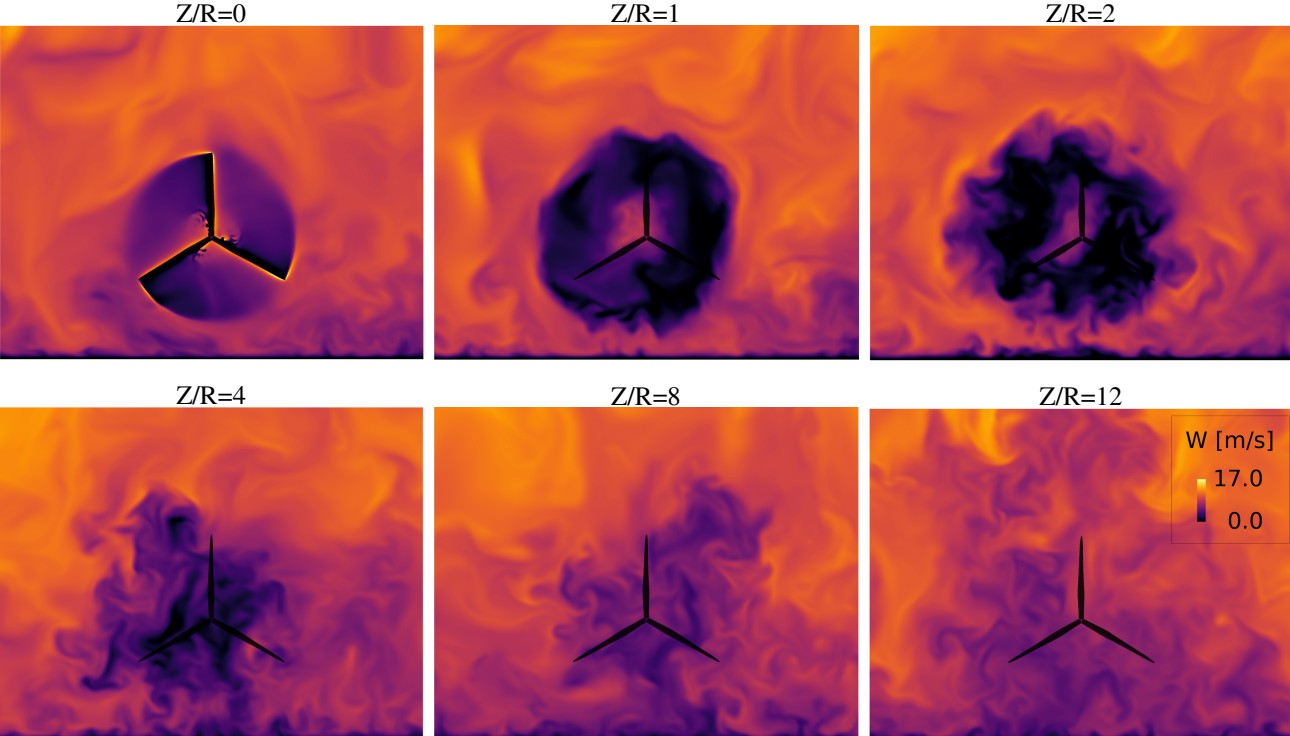

**Figure 10.** Velocity fields downstream of rotor.

### 3.2.1 Impact of turbulence model

To study the impact of the presented turbulence model on the flow, simulations with the hybrid LES/IDDES blending enabled
along with pure IDDES simulations are conducted. In the pure IDDES simulation, a slip wall condition is used on the terrain surface, contrary to the log-law used for the LES/IDDES hybrid model. Simulations with and without the rotor present were simulated. In the empty setup, the hybrid model acts as a pure Deardorff LES model, as no blending region is defined. For all simulations, inflow is interpolated from the LES precursor planes to ensure identical inlet conditions. In the simulations comparing turbulence models, only the CFD code has been used, meaning that no flexibility of the blades is considered.

Firstly, the empty setups are presented in Figure 11 showing the velocity component $W$ at the center planes, for the simulations with Deardorff and IDDES turbulence modelling at the same time instance. From the planes, two instantaneous profiles are extracted (dashed lines), which are shown in Figure 12. Both simulations show very comparable results. As seen, the velocity profiles, extracted 96m from the inlet, are practically identical, while a small discrepancy is seen further downstream in the domain as a result of changing the turbulence and wall models.

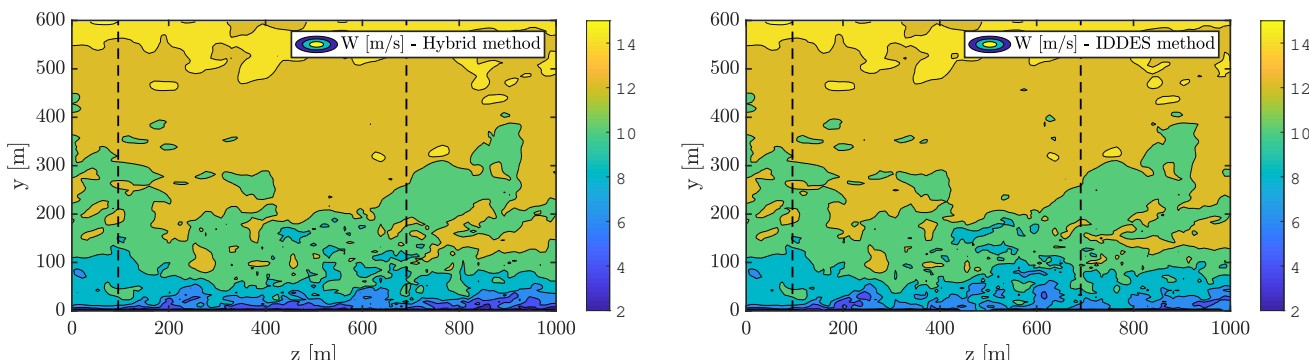

**Figure 11.** Instantaneous flow field ($W$ component) for hybrid (left) and IDDES (right) simulations

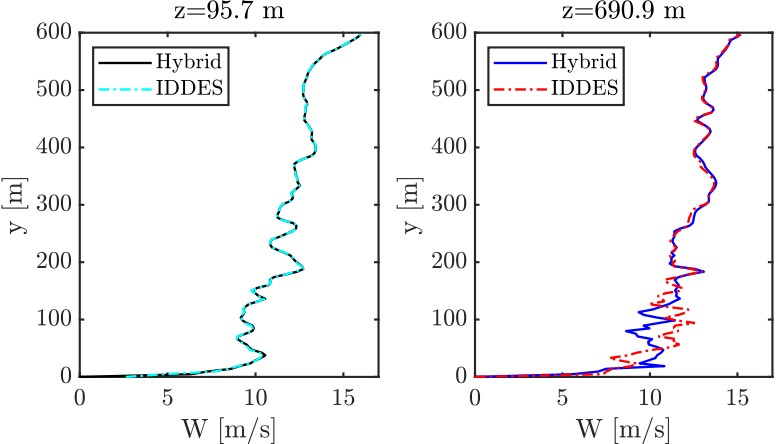

**Figure 12.** Instantaneous velocity profiles extracted from planes close to the inlet (96m from inlet) and far downstream (691m from inlet) at time=129.6 seconds, see dashed lines of Figure 11.

Stiff simulations covering 35 rotor revolutions were conducted with the two turbulence models, including the rotor in the simulations. Mean and standard deviations of azimuthal forces of the final 15 revolutions at two blade sections, near mid and near tip, are presented in Figure 13. Only slight differences are seen in both mean and standard deviations between the two models, aligning well with what is seen in the empty domain simulations. As the incoming flow is not altered significantly and the turbulence model near the blade is IDDES in both simulations, the forces are expected to be similar as well.





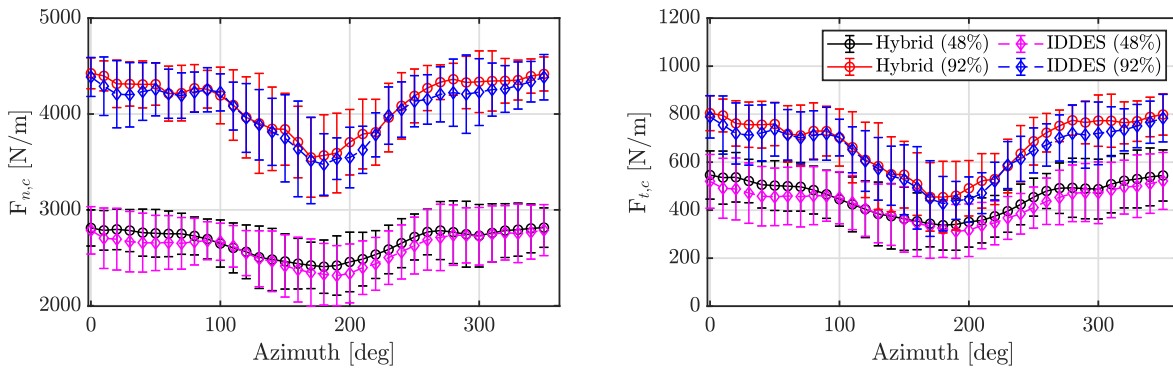

**Figure 13.** Normal and tangential forces at 48% and 92% blade length using hybrid or IDDES turbulence model. Temporal means and standard deviations based on the final 15 revolutions

### 3.2.2 Impact of flexibility

To study the effect of the rotor flexibility, FSI simulations of flexible and stiff setups were performed. First, 35 revolutions were simulated through pure CFD, as presented before, followed by 25 revolutions with the FSI coupling enabled, see Figure 3 for the FSI simulation process.

The following results are obtained using the hybrid turbulence model only, but similar results would be expected for pure IDDES simulations, based on the aforementioned findings. The effect of including the blade flexibility is assessed through the resulting blade displacements, torsion and the rotor loading.

Figure 14 depicts the tip displacement flap- and edgewise along with the resulting blade torsion at 60.1% and 95.3% blade length. The tip displacement in flapwise direction is ≈6% of the blade length with fluctuations up to ≈1% due to the turbulent flow. Edgewise displacements are low and dominated by gravity, seen in the more regular pattern and low standard deviation.

Blade torsion is quite low as well, with less than 0.5° near the tip, increasing the angle of attack.





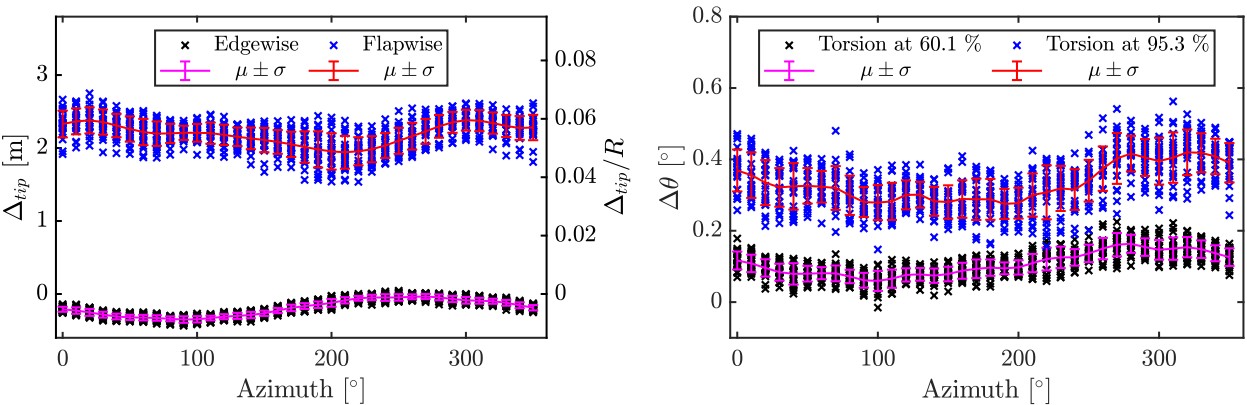

**Figure 14.** Right: Tip displacements flap- and edgewise. Left: Blade torsional deformation at 60.1% and 95.3% blade length

In Figure 15, the integrated rotor thrust and torque are depicted, showing that in general only slight differences are seen by including flexibility. This is seen, with an increase in thrust of 1-5% while no significant change is seen in torque, other than a slight decrease in fluctuation amplitudes when including flexibility. As seen, the turbulent inflow results in a much higher
thrust/torque variation than seen from considering flexibility. This is seen in both large time scales with the low frequency fluctuations, along with small scale fluctuations within the individual revolutions.

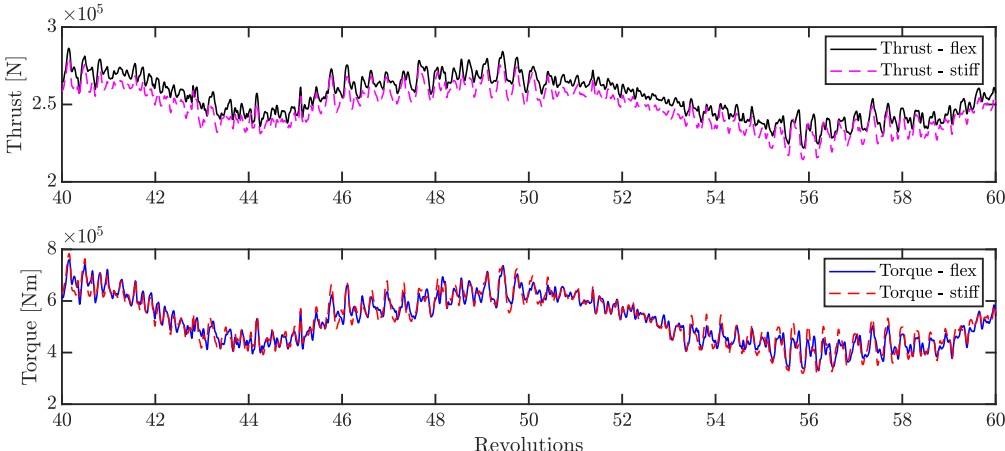

**Figure 15.** Integrated thrust and torque for stiff and flexible configurations

As mentioned, some differences are present in the simulation setup compared to the DanAero field experiment, being the omission of yaw, tilt and tower along with the higher turbulence intensity of the generated flow.

Despite this disclaimer, the resulting forces at four sections of the blade are depicted in Figure 16, showing the mean
azimuthal pressure forces normal and tangential to chord for both flexible and stiff simulations along with the DanAero measurements. As seen, the forces agree well between the two simulations and the measurements with main differences being





the lack of tower shadow at the inner sections. The standard deviation of forces is seen to be higher in simulations than for measurements, which is expected as the turbulence intensity of the sampled flow is higher than measurements as seen in Figure 7.

The impact of including flexibility is quite small, and general observations are that normal and tangential forces respectively slightly increase and decrease when considering the flexibility of the rotor. This is expected for the NM80 rotor, which is quite stiff compared to modern wind turbines. The standard deviations of the forces due to turbulence, are much higher than the difference between mean forces of stiff and flexible simulations. This shows that including turbulent inflow is more important than including flexibility, at least in the present rotor/flow case. In the simulations including the flexibility the standard devi-

ations of the normal forces are up to 10% of the mean near the tip and 15% near the root. For tangential forces this is even higher with 24% near the tip and 38% near the root.



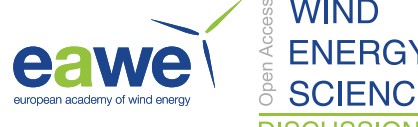

**Figure 16.** Normal and tangential forces for stiff and flexible simulations along with DanAero measurements. Sections at 33%, 48%, 76%, 92% blade length respectively





Spectral analysis of the resulting normal and tangential force signals at the 76% blade length section are presented in Figure 17 (left) showing the power spectral densities (PSD) using the Welsh estimate. As seen, both stiff and flexible simulations result in similar PSDs, with the main difference being the peak at the first edgewise frequency seen in the flexible signal. The majority of energy is found in the rotation frequency, 1P, and its harmonics. This is also the case when looking at the PSD of the tip displacement in flap and edgewise direction. Here, it is again the rotation frequency and its harmonics that dominate, along with a peak of the first edgewise mode.

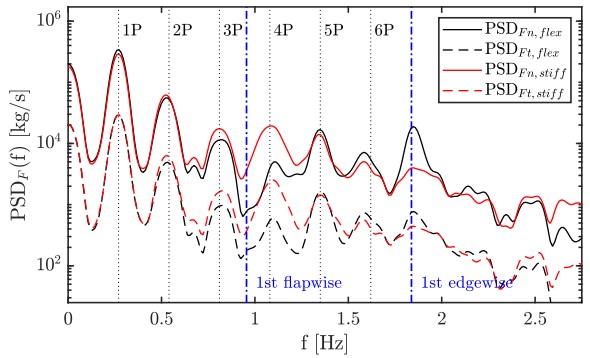 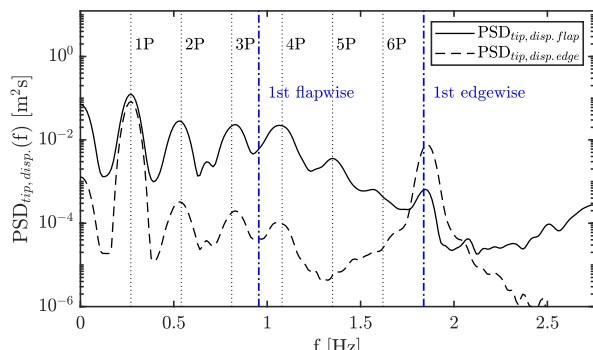

**Figure 17.** Left: PSD of normal and tangential forces at 78% blade length for both stiff and flexible simulations. Right: PSD of tip displacement in flap and edgewise direction

## 4 Conclusions

This study investigates the phenomenon of aero-elasticity of wind turbines placed in atmospheric flow conditions, by means of high fidelity numerical methods. Fluid-structure interaction (FSI) simulations of a 2.3MW wind turbine rotor have been conducted using a novel turbulence model, blending the Deardorff large eddy simulation (LES) model for atmospheric flows with the improved delayed detached eddy simulation (IDDES) model for the separated flow near the rotor boundary. Precursor simulations were conducted on a large domain in order to assure sampling of realistic turbulent atmospheric boundary layer (ABL) flow, matching well with the DanAero measurements, for the successor simulations.

As a first study, the hybrid model was compared to the pure IDDES turbulence model, by CFD successor simulations of the turbulent ABL inflow with and without the rotor present. In empty simulations, this corresponded to a comparison between pure Deardorff LES and pure IDDES, while for rotor simulations the hybrid model used both Deardorff LES for the domain flow and IDDES for the near rotor flow. It was found that there was no significant difference in the flow nor rotor loading between the two methods, likely due to the short domain considered and assumptions omitting Coriolis force and temperature effects.

Secondly, FSI simulations have been conducted by coupling the CFD simulations to a structural solver. It was found that for the specific rotor, which is relatively stiff compared to modern turbines, only small impact was found by considering the



flexibility of the blades. A general increase of $\approx$1-5% in total thrust was found, while the power producing torque was close to identical for stiff and flexible simulations.

Inflow turbulence on the other hand has a large influence on the rotor loading, with standard deviations as high as 15% of the mean for normal forces and even higher tangentially. This emphasizes the importance of correct modelling of inflow turbulence.

## 5   Future studies

As it was shown in the present study, the developed hybrid turbulence model resulted in practically identical loading of the rotor
as the IDDES model alone. Relevant future studies would be to investigate when this is not the case. This could for instance be simulations including stable/unstable stratification and/or Coriolis force. Here, the IDDES model will probably be insufficient to capture the effects, as the model is calibrated for aerodynamics mainly and not ABL flows. Longer domains with multiple rotors could also be relevant, as there is time and distance for the two turbulence models to develop the flow differently.

A relevant future study would likewise be to compare the method to more efficient BEM based aerodynamics solvers with
the precursor turbulence as input. Here, the CFD-based results could, if needed, be used to correct airfoil polars and calibrate the many correction models needed by BEM solvers to consider e.g. tip loss effects, dynamic inflow and dynamic stall.

In terms of FSI, it would be natural to investigate more recent/future turbine designs, which are larger and much more flexible than the considered NM80 rotor. These rotors are in higher risk of instability phenomena, and operate in a larger part of the atmospheric boundary layer.

*Code and data availability.*   The codes used to conduct the presented simulations are licensed and not publicly available. Data can be shared upon request

*Author contributions.*   CG conducted all simulations, including grid generation, and implemented the presented turbulence model in the CFD code. Additionally CG did the analysis of the results and has been the main writer of the paper. NNS assisted with expertise in the code development and CFD setup, along with planning the study and analysis. SGH has supported in especially FSI setup along with planning
and analysis. NT supported the development and implementation of the turbulence model. FZ assisted in the overset grid setup, and did development on the CFD code to accelerate simulations. Further, all authors contributed in writing and editing this paper.

*Competing interests.*   The authors declare that there have been no conflicts of interest during this study



*Acknowledgements.* This study was funded by DTU Wind Energy, through the PhD project "Fluid-structure Interaction of Wind Turbines in Atmospheric Flow". The DanAero projects, from which experimental data was obtained, were funded partly by the Danish Energy Authorities, (EFP2007. Journal nr.: 33033-0074 and EUDP 2009-II. Journal nr. 64009-0258) and partly by eigenfunding from the project partners.





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
