# Peer review of "Wind turbines in atmospheric flow - Fluid-structure interaction simulations with hybrid turbulence modelling"

_Wind Energy Science, 2020_

## Referee Comment (RC1) · Anonymous Referee #1 · 5 Jan 2021

The manuscript introduces a new method to perform detailed simulations of the Fluid Structure interaction of wind turbines in an atmospheric flow. The work is novel and relevant for Wind Energy Sciences readers. The manuscript is well very written and easy to read. The used simulation method is very involved and builds upon previous work by the authors. Overall, I believe the authors provide a good summary of the developed simulation method and refer to their previous work when appropriate. As the authors write modeling choices are unavoidable and appropriate validations for the case under consideration are presented. I believe the simulations are performed carefully. After reading the manuscript I have several comments / questions, see below, which I would like the authors to comment on.

[Figure]

* The last part of the introduction, starting at "However, it is expected" is somewhat speculative as this is not actually shown in the manuscript and should be removed. * Line 87 (and other places): "The study is a continuation of (Grinderslev et al.)". This reference is not available to the referee and hence it can not be checked how this work is related to what is presented there. A short paragraph describing the connection between both works could be beneficial. * Figure 1: Can you include a scale (meters in the graph). Also where is the blend factor Fh defined? * line 271: in line 316 it is mentioned that no tower and nacelle are considered, so I am unsure how to interpret the statements here. * line 272: "for the initialization" ==> arguably this is part of the simulation method part (section 2.4) * line 290: directions of U,V,W should be defined * line.321: Can you provide a reference or additional explanation on this. * line 328/329: "The disc and rotor grids are similar to the setup used in (Grinderslev et al.), however in this study the background grid has changed to be suitable for LES." ==> Can you please clarify what is meant by making the grid suitable for LES? * Figure 4: the blue grid seems to extend further above the blades than below. Is there a reason for this, or my mistake in interpreting the figure? Is the "bending" of the blue region on effect of the bending of the blades? Just as figure 1 can you include a scale? * Figure 6: panel a yellow line is for 77 meters and in panel b for 93 meters. Please check. * Figure 10: please include a color scale. * Figure 12: This figure shows a snapshot, which shows that the instantaneous realization is different in the region up to 150 meters, i.e. the region where the turbine is roughly located. Did you check whether the turbulence intensity is affected by changing the model? * Figure 14: Do the crosses represent instantaneous realizations? * Figure 16: Do you know up to what degree omitting the tower and nacelle could influence the simulation result? * line 429: Here it is concluded that the domain may be too short. However, I could not find the corresponding discussion in the main text on which this conclusion is based. * charter 5: Some comments, for example why IDDES would be likely be insufficient for some cases, would benefit from some additional explanation.

[Figure]

[Figure]

---

## Referee Comment (RC2) · Anonymous Referee #2 · 14 Jan 2021

General comments.

The manuscript presents high fidelity rotor-resolved numerical simulations under neutral atmospheric conditions. A new hybrid turbulence model between is presented and results are compared to the use of IDDES only. Simulations are performed for both rigid and flexible blades. The work is very interesting and highly relevant to the field. The manuscript is well-written and rather complete. The referee thinks the paper can be accepted for publication after minor revisions, see below.

Specific comments.

- Neutral conditions are considered here. The authors mention the fact that stratifications should also be done as future work and is likely to impact on the outcome of the turbulence model. The manuscript would benefit from more explanations about this, including the challenges of modelling stable and unstable conditions with the presented CFD model.

- Some information on the model settings are missing for the reader to fully reproduce these simulations (or at least not obviously presented). Please make sure all the input parameters (e.g. of the turbulence model, blade structural parameters) are specified in the text, in an appendix if needed. The overall Reynolds number could also be mentioned in the text.

- L260: In this study, the LES length scale is fixed to the background mesh so that it doesn't change with mesh refinement. Can you comment on the impact that this has on the cut-off of scales, e.g. in the turbulent kinetic energy spectrum?

- Tower and nacelle are omitted from the CFD results but they will impact on the near wake. Please comment on this.

- L326: Can you specify more precisely at what distance the hole of the disc mesh should be from the surface?

- L340: 1000sec are considered for the post-processing, Can you add also the corresponding time period in rotor revolutions?

- Adding details of the computational cost (resources used) would be useful.

Minor corrections.

- L122: word "for" is duplicated

- L170: equal -> equal to

---

## Author Comment (AC1) · 12 Feb 2021

**Reviewer 1**

Black bold marks reviewers comments, blue text marks authors' answers to comments

**The manuscript introduces a new method to perform detailed simulations of the Fluid Structure interaction of wind turbines in an atmospheric flow. The work is novel and relevant for Wind Energy Sciences readers. The manuscript is well very written and easy to read. The used simulation method is very involved and builds upon previous work by the authors. Overall, I believe the authors provide a good summary of the developed simulation method and refer to their previous work when appropriate. As the authors write modeling choices are unavoidable and appropriate validations for the case under consideration are presented. I believe the simulations are performed carefully. After reading the manuscript I have several comments / questions, see below, which I would like the authors to comment on.**

The authors would like to thank the reviewer for the great comments and suggestions. The points made are very good and will be considered in the revised manuscript. Please see answers at the comments below, and see the attached revised version of the manuscript with marked changes of the text.

**- The last part of the introduction, starting at "However, it is expected" is somewhat speculative as this is not actually shown in the manuscript and should be removed.**

The sentence has been removed from the abstract.

**- Line 87 (and other places): "The study is a continuation of (Grinderslev et al.)". This reference is not available to the referee and hence it can not be checked how this work is related to what is presented there. A short paragraph describing the connection between both works could be beneficial.**

This is unfortunate, as the paper is still awaiting publication in another journal. The major connection is that the mentioned reference also considers FSI on the NM80 rotor in a similar flow scenario along with two other flow scenarios. The main difference is the turbulence modelling which in (Grinderslev et al., 2021) is based on URANS, so without any modelling of turbulent fluctuations, meaning only ensemble averaged effects are considered. The novelty of the present paper lies in the turbulence modelling and the combination of all the aspects of: overset grids, FSI coupling and high fidelity turbulence modelling.

The following was added/changed in the text

"*The study is a continuation of Grinderslev et al. (2021), where FSI of the NM80 rotor was studied in various flow scenarios with sheared and yawed, however laminar, inflow using URANS turbulence modelling. The flow scenario studied here resembles one of these studied laminar scenarios, however with turbulent inflow through the novel hybrid turbulence model and an adjusted grid setup.*"

**- Figure 1: Can you include a scale (meters in the graph). Also where is the blend factor Fh defined?**

A scale of the blade length has been added to give a better impression on the spatial scale.

"Definition" has been removed from the caption, as the figure more visualizes the blending regions. Further, the text before has been clarified, to explain that the majority of the blending happens in the range 8m ±2m from the surface. The definition of the blend factor is described in Eq. 8.

**- Line 271: in line 316 it is mentioned that no tower and nacelle are considered, so I am unsure how to interpret the statements here.**

The reason is that the tower, shaft, hub and nacelle are present in the HAWC2 structural model to assemble the rotor however not in the CFD setup. For this reason, they are considered rigid and take no part in the simulations.

The sentence was rewritten:

*"As only the rotor is modelled in CFD, only blade flexibility is considered as well. **This means that tower, nacelle, shaft and hub are not active parts of the HAWC2 simulations.**"*

**- Line 272: "for the initialization" ==> arguably this is part of the simulation method part (section 2.4)**

This is true. The sentence has been removed, as a corresponding sentence is present in the simulation method part.

**- Line 290: directions of U,V,W should be defined.**

U velocity is horizontal and perpendicular to the mean flow direction, V is vertical and perpendicular to the mean flow direction and W is the mean flow direction .

This has been added to the text.

**- Line.321: Can you provide a reference or additional explanation on this.**

This is purely a measure to simplify and speed up the holecutting process for the overset grid method. Instead of doing holecutting routines for every timestep as the rotor moves, the manually cut holes in the disc mesh are moving along with the rotor mesh. This means that the receiver and donor cell ID's do not change during the simulations, as to why the search between these two grids is only necessary once.

A benefit from this is that one can force any deformation in the rotor mesh to happen within the mesh located inside the manually cut holes. By this, the donor cells of the rotor mesh will not deform nor move relatively, which removes the need of re-searching for receiver/donor cells and re-calculating interpolations coefficients. It is purely a matter of speeding up simulations.

The disc and background mesh, however, need updates for each time steps as the disc moves and the background is static. As the geometry here is much simpler, the holes in the background mesh is easily calculated through a simple analytical cylindric shape hole, which is predefined and thereby fast and efficient.

The text has been updated with more sentences to explain it more clearly. For instance the following was added:

*"In the present setup the holes are 17m wide in the rotor axis direction, where the main deformation is present, located with the undeformed rotor in the center. Displacements are propagated to the volume mesh, such that points within the inner 15% of the grid curve length normal to the surface are moved as solid body motion to ensure no change of quality of the inner cells resolving the high gradient flow. Further out, from 15% to 40% the volume blend factor is linearly decaying from 1 to 0, such that points from 40% grid curve length and out are unchanged to avoid changes to donor cells.*

*Note that communication tables and hole-cutting still need update between disc mesh and background mesh as the latter is static."*

**- Line 328/329: "The disc and rotor grids are similar to the setup used in (Grinderslev et al.), however in this study the background grid has changed to be suitable for LES." ==> Can you please clarify what is meant by making the grid suitable for LES?**

The background grid in this study is rectangular with rectangular cells, which have low stretching in the focus area (area leading up to and behind the rotor). In the mentioned previous study, a semi-cylindrical background mesh was used, where cells where less uniform due to the circular shape. This was of low importance for the URANS turbulence, however, as the Deardorff LES model is directly sensitive to mesh cell sizes, the semi-cylindrical mesh was deemed unfit for use in this study.

A comment has been added to the text saying

*"…suitable for LES simulations by using rectangular cells with low cell stretching in the area of focus"*

**- Figure 4: the blue grid seems to extend further above the blades than below. Is there a reason for this, or my mistake in interpreting the figure? Is the "bending" of the blue region on effect of the bending of the blades? Just as figure 1 can you include a scale?**

This is an effect of the figure looking at the rotor from the side, making the grid of the lower blades look shorter, as the blades are not positioned vertically. If you look closer to the right figure, one can see that the mesh is identical for the three blades.
The reason for the bending shapes is indeed the effect of the pre-bending of the blade. The mesh is "grown" directly from the surface mesh using the hyperbolic Hypgrid3D mesher, and no desired outer shape was defined.

A scale was added, by displaying the hub height distance on the left figure.

**- Figure 6: panel a yellow line is for 77 meters and in panel b for 93 meters. Please check.**

This is a mistake .The right label is 93m, and the figure has been corrected.

**- Figure 10: please include a color scale.**

The color scale on the lower figure to the right has been emphasized by adding a white background to the color bar.

**- Figure 12: This figure shows a snapshot, which shows that the instantaneous realization is different in the region up to 150 meters, i.e. the region where the turbine is roughly located. Did you check whether the turbulence intensity is affected by changing the model?**

As shown below, there is a slight change when moving far down the domain, but in this case the significance is not high. This figure has been added to the paper.

[Figure]

**- Figure 14: Do the crosses represent instantaneous realizations?**

Yes they do indeed. This information has been added to the caption of the figure.

**- Figure 16: Do you know up to what degree omitting the tower and nacelle could influence the simulation result?**

We have not tested this ourselves, but studies by e.g. (Guma et al. 2020), suggest that the influence is quite small for this rotor and very local to the part of the azimuth when the blade passes the tower (160-200 deg). In the study by Guma et. al, the authors compared a full turbine configuration with a rotor only configuration. The effect was mainly seen in the fatigue damage equivalent loading (DEL), but the section forces were in general similar.

The omission of the tower mainly influences the inboard sections of the blade. This is seen in Figure 16, top row, showing the forces at 33% blade length. The CFD results are nearly flat, while the measurements drop a bit when passing the tower.  For the sections further outboard, the CFD results show similar trends as the measurements, as a result of the sheared inflow being dominant.

For larger and more flexible rotors, as considered in (Santo et al. 2020) the effect might be more significant, as the force dip from the tower shadow will make the blade deformation fluctuate more. However, the load effect still is quite small in the study (Santo et al. 2020).

The nacelle is deemed to be of low interest as well, as the loads around the most inner part of the blades are low.

The following was added to the document when introducing the CFD mesh:
*"In the present setup, only the rotor is considered, omitting the tower, hub and nacelle, with a total of three overlapping mesh groups, see Figure 4.* **The omission of these elements is expected have a minor effect, supported by the study (Guma et al.,(2021)) studying the same rotor represented both as a rotor only and as a full turbine.**

The comment on Figure 16 has been clarified also.

*"As seen, the forces agree well between the two simulations and the measurements with main differences being the lack of tower shadow at the inner sections*, **resulting in a drop of loading in measurements not seen in simulations (see Figure 16 top row)***"

**- Line 429: Here it is concluded that the domain may be too short. However, I could not find the corresponding discussion in the main text on which this conclusion is based.**

The domain is not deemed too short for the purpose of the study conducted here, however it is too short to clarify large differences in the resulting flow behavior between the two turbulence models. This is mainly based on the flow starting to vary between the IDDES and hybrid changing in the back part of the domain as depicted in figures 11 and 12. For longer domains, for instance when considering more than one rotor, this effect would likely build up.

In the text commenting on figures 11 and 12 (line 375) the following was added:

*While both turbulence models perform close to identically for the present single turbine study, it could be speculated that this would not be the case when considering larger domains, for instance when studying multiple turbines at once.*

*In that case, the differences in turbulence and wall modelling will likely result in different flow fields, due to the longer distances covered. Considering temperature effects on the flow might likewise reveal differences between the turbulence models in terms of temperature flux, along with the Deardorff model considering the stability in the mixing length scale, which is not the case in the IDDES model.*

**- Chapter 5: Some comments, for example why IDDES would be likely be insufficient for some cases, would benefit from some additional explanation.**

The reason for the assumption is mainly that IDDES is based and calibrated for separated flows only, whereas the LES model is calibrated to ABL flows. This is for instance evident in the case of stable stratifications, where the length scale is altered in the Deardorff LES model, based on the temperature gradient. This is not handled in IDDES.

There are differences in how the two models consider temperature as well in the energy equations, as there is no buoyancy term in the original k-omega model by Menter. In Ellipsys3D the buoyancy is considered through $\tau_{\theta w,LES}$ and $\tau_{\theta w,DES}$ which are modelled with different weights of the turbulent viscosity.

The following comments were added to the text:

*"Here, the IDDES model will probably be insufficient to capture the effects, as the model is calibrated for aerodynamics mainly and not ABL flows. The Deardorff LES model, however, is calibrated for such flows and the mixing length scale depends on the stratification, as it is reduced for stable cases. The two models also model temperature effects differently as the flux, $\tau_{\theta w}$, is based on different weights of the turbulent viscosity."*

**Reviewer 2**

Black bold marks reviewers comments, blue text marks authors' answers to comments

**General comments. The manuscript presents high fidelity rotor-resolved numerical simulations under neutral atmospheric conditions. A new hybrid turbulence model between is presented and results are compared to the use of IDDES only. Simulations are performed for both rigid and flexible blades. The work is very interesting and highly relevant to the field. The manuscript is well-written and rather complete. The referee thinks the paper can be accepted for publication after minor revisions, see below**

The authors would like to thank the reviewer for revising the manuscript and providing very useful comments and corrections which will be considered in the revised manuscript. Answers to comments and revision to text are written below in connection to the reviewer comments. See also the attached revised version of the manuscript with marked changes of the text.

**Specific comments:**

**- Neutral conditions are considered here. The authors mention the fact that stratifications should also be done as future work and is likely to impact on the outcome of the turbulence model. The manuscript would benefit from more explanations about this, including the challenges of modelling stable and unstable conditions with the presented CFD model.**

The presented hybrid model, will in its nature act as the general Deardorff LES model for stratified flows, assuming that the IDDES region is small and only present near the considered object.
The Deardorff model is calibrated for stratified flows, and has a reduction to the mixing length scale when in stable conditions. This is one of the reasons that it is assumed to perform better than the original IDDES, as this model is calibrated for separated flows and has no specific corrections depending on stratification.

The following was added to the text of chapter 5:

*"Here, the IDDES model will probably be insufficient to capture the effects, as the model is calibrated for aerodynamics mainly and not ABL flows. The Deardorff LES model, however, is calibrated for such flows and the mixing length scale depends on the stratification, as it is reduced for stable cases. The two models also model temperature effects differently as the flux, $\tau\theta w$, is based on different weights of the turbulent viscosity."*

**- Some information on the model settings are missing for the reader to fully reproduce these simulations (or at least not obviously presented). Please make sure all the input parameters (e.g. of the turbulence model, blade structural parameters) are specified in the text, in an appendix if needed. The overall Reynolds number could also be mentioned in the text.**

As the considered turbine design is not publicly available, the structural model can unfortunately not be provided. However, it is referenced from (Bak et al., 2013), for those who are allowed access, (an NDA might need to be signed).

The authors believe that the turbulence modeling is described comprehensively in section 2.2, with explanations of the new model proposed, and with references to the background models used.

Physics, time steps, convective schemes used and velocity/pressure algorithm for CFD simulations are all described in section 2.3.1.

The Reynolds number has been found to ≈6M in the majority of the blade, based on the rotational speed and the hub height velocity. This has been added to the section 2.3.1:

*"The rotation speed of the rotor is constant to 16.2rpm resulting in an effective Reynolds number of ≈6M along the majority of the blade for the studied flow case."*

Grid setups including boundary conditions are described in section 2.5.

**- L260: In this study, the LES length scale is fixed to the background mesh so that it doesn't change with mesh refinement. Can you comment on the impact that this has on the cut-off of scales, e.g. in the turbulent kinetic energy spectrum?**

In the present case, where mesh is refined in a small region only, the effect of fixing the LES length scale to the background grid has little effect. The turbulence in the incoming flow, which is based on the background grid length scale, will not have time to dissolve into smaller scales in the little distance before going to the IDDES region, where the length scale is based on the actual grid size.

More importantly, is likely the distance behind the rotor, where small scale separated flow is present. In this study, it has been assumed that the distance of the IDDES region is sufficiently long for resolving these small scales of the flow and the background grid resolution is low enough for resolving the wake further away. This could be an interesting parameter study, as a longer IDDES region (and longer rotor and/or disc mesh) would enable the IDDES model in a larger region. In this study, the background grid is still quite fine and the fine background cells are present until 6D behind the rotor.

In the preparation of the present study, and for previous tests of the model alternative strategies were tested as well.
It has for instance been attempted to increase the size of the IDDES region, such that all overset grid refinements were within the IDDES region.
By this, the IDDES length scale was used, but as this is only present in the dissipation term of the IDDES model (and not also directly in the viscosity term as the Deardorff) the sudden jump of viscosity is avoided. Using this alternative method gave the same results as the strategy used here limiting the LES length scale. It was chosen to go for the background grid limiter, as this allows as large a portion of the ABL LES model in the flow, which for other stratifications might have an impact. However, if the background grid is of coarser resolution in a study, this approach might not be ideal.

The following was added to the text:

*"The length scale limit of the LES region, caps the frequency range of the resolved turbulent kinetic energy to the background grid resolution.*

*Close to the rotor, however, the small scale detached flow is still captured through the IDDES model. For studies, with long distances from refinement to object or larger resolution differences between background and overlapping sub-grids, this strategy would likely not be optimal due to the capping of resolved frequencies being based on an unnecessarily large grid size."*

**- Tower and nacelle are omitted from the CFD results but they will impact on the near wake. Please comment on this.**

Based on a study by Guma et al. (Guma et al., 2020) (https://wes.copernicus.org/articles/6/93/2021/wes-6-93-2021.html ) looking at the same rotor, the influence is low on the sectional loads and displacements. However, there is an effect in fatigue loads as the drop in forces when passing the tower is omitted.

In the present study effects in loading is mainly seen inboard on the blades (see the azimuthal loading at 33%), which have a drop in forces when passing the tower in measurements not seen in simulations. Further out, the shear is dominant in the force variation.

The following was added to the document when introducing the CFD mesh:

*"In the present setup, only the rotor is considered, omitting the tower, hub and nacelle, with a total of three overlapping mesh groups, see Figure 4. **The omission of these elements is expected to show little effect, based on the study of Guma et al., studying the same rotor represented as a rotor only and a full turbine.**"*

The comment on Figure 16 has been clarified also.

*"As seen, the forces agree well between the two simulations and the measurements with main differences being the lack of tower shadow at the inner sections**, resulting in a drop of loading in measurements not seen in simulations (see Figure 16 top row)**"*

**- L326: Can you specify more precisely at what distance the hole of the disc mesh should be from the surface?**

The distance and smearing of deformation to the volume grid is something, which is tuned for the specific setup, as it is naturally dependent on the degree of deformation.
The smearing blend was done linearly in the range of 15-40% of the length of the grid line of the rotor grid normal to the surface. This means that within the first 15% of the grid line length from the surface, the entire deformation is propagated, while between 15% to 40% the blend factor linearly goes from 1 to 0, such that after 40% no deformation is added to the mesh points. This made sure that the blend factor was 0 in the location of the hole of the background grid such that donor cells were not altered.
The manually cut holes are approximately 17m wide (along the rotor axis), with the un-deformed rotor placed around the center, leaving around 8.5m for deformation of the rotor mesh.

The following was added to the text:
*"In the present setup the holes are 17m wide in the rotor axis direction, where the main deformation is present, located with the undeformed rotor in the center. Displacements are propagated to the volume mesh, such that points within the inner 15% of the grid curve length normal to the surface are moved as solid body motion to ensure no change of quality of the inner cells resolving the high gradient flow. Further out, from 15% to 40% the volume blend factor is linearly decaying from 1 to 0, such that points from 40% grid curve length and out are unchanged to avoid changes to donor cells.*

*Note that communication tables and hole-cutting still need update between disc mesh and background mesh as the latter is static."*

**- L340: 1000sec are considered for the post-processing, Can you add also the corresponding time period in rotor revolutions?**

1000 seconds correspond to ≈270 revolutions. However, note that the 1000 seconds were only used for the postprocessing of the precursor simulation (i.e. for spectras and statistics). For the successor FSI simulations a total of 60 revolutions were simulated, meaning that only part of the sampled flow was used. The longer period of the precursor was to ensure a sufficient sample size, as well as it was not know in advance how many seconds the successor simulations would take to converge.

The sentence was clarified:

"*A total of 9750 seconds were simulated for the precursor simulation, of which the final 1000 seconds* **(equivalent to ≈270 rotor revolutions) were sampled for statistical post processing,** *in a period where the developed flow profile sufficiently matched the desired profile.*"

**- Adding details of the computational cost (resources used) would be useful.**

The following was added to the text:

Sec 3.1: *"The full precursor simulation was conducted on 1728 AMD EPYC 2.9GHz  processors on the computer cluster of DTU, and lasted ≈45 hours."*

Sec 3.2: *"For the initial phase 1, simulations were conducted on 1189 AMD EPYC 2.9GHz processors, while coupled simulations (phases 3 and 4) were conducted on 793 Intel Xeon 2.8 GHz processors. The initialization simulations took in the order of ≈26 wall clock hours, while the coupled phases lasted for ≈180 wall clock hours per simulation."*

**Minor corrections:**

These have been corrected

- L122: word "for" is duplicated

- L170: equal -> equal to

[revised manuscript text omitted]
 various flow scenarios with sheared and yawed, however laminar, inflow using URANS turbulence modelling. The flow scenario studied here resembles one of these studied laminar scenarios, however with turbulent inflow through the novel hybrid turbulence model and an adjusted grid setup.

**2 Methodology**

95 In this section, the computational solvers are presented along with the simulation strategies such as FSI framework and precursor simulations. Further, the  participating turbulence models will be introduced, to prepare for the discussion of the hybrid model. Finally, the computational grids used in the study are described along with the chosen simulation parameters.

**2.1 Numerical methods**

**100 2.1.1 Flow solver**

To solve the fluid flow, the DTU inhouse CFD code EllipSys3D (Michelsen, 1992, 1994; Sørensen, 1995) is used. The code solves the incompressible Navier-Stokes equations in structured curvilinear coordinates using the finite volume method with a collocated grid arrangement. The code is parallel and highly scalable using the message passing interface (MPI) and multi-block decomposition, the multi-grid method and grid sequencing. EllipSys3D has multiple convective schemes implemented,

105 such as central difference (CDS), second order upwind (SUDS) and quadratic upstream interpolation for convective kinematics (QUICK). For solution of the pressure correction equation, various algorithms are implemented such as PISO, SIMPLE, SIMPLEC and variations hereof. Rhie-Chow interpolation is used to avoid odd/even pressure decoupling. Overset capabilities, including grid hole-cutting are implemented internally in the code (Zahle et al., 2009).

Several turbulence models are implemented such as two equation Reynolds Averaged Navier Stokes (RANS) models, $k - \epsilon$

110 and $k - \omega$ among others, hybrid models like detached eddy simulations (DES), delayed DES (DDES), improved DDES (IDDES) and multiple large eddy simulations (LES) models. In addition to these, a hybrid version of the LES and the IDDES model will be presented in this paper.

For FSI simulations, the deformation of grids is handled through a moving mesh method with a  blend factor. The surface displacement is propagated along the grid lines normal

115 to the surface, with a blend factor gradually diminishing to zero with increasing distance to the blade surface. The blending can be either linear in the distance to the blade surface or based on a tanh function. 
[revised manuscript text omitted]

**2.3.2**

**Turbulence blending**

To enable the hybrid turbulence modelling, a blending region must be defined. As mentioned in Section 2.2.3, a sudden grid refinement will create a sudden length scale change, and thereby, if in the Deardorff LES region, a sudden change of turbulent viscosity. In the present setup with overset grids, it is therefore chosen to avoid the viscosity "jump" by keeping the LES length scale $\Delta_{LES}$ to the background grid value. By this, the refinement does not change the dissipation length scale nor the viscosity. Near the rotor however, an IDDES  region is prescribed depending on the wall distance.  In this region, the refined mesh impacts the turbulent dissipation through $\Delta_{DES}$, as usual.

The length scale limit of the LES region, caps the frequency range of the resolved turbulent kinetic energy to the background
grid resolution. Close to the rotor, however, the small scale detached flow is still captured through the IDDES model. For
studies, with long distances from refinement to object or larger resolution differences between background and overlapping
sub-grids, this strategy would likely not be optimal due to the capping of resolved frequencies being based on a unnecessarily
large grid size.

In the present setup the blending between LES and IDDES happens with midpoint ($F_h = 0.5$) 8m from the surface with the
majority of the blend happening over  $\pm$2m distance from this point to ensure a smooth transfer from LES to
IDDES, see Figure 1. ~~In the IDDES region, the refined mesh impacts the turbulent dissipation through $\Delta_{DES}$, as usual. By
this, the small scale detached flow is still captured close to the rotor.~~

[Figure]

**Figure 1.** Blend factor, $F_h$. Red: IDDES region, Blue: Deardorff LES region. Isosurface of blend factor $F_h$=0.9 ($\approx$6m normal
from surface)

**2.3.3 HAWC2 model**

[revised manuscript text omitted]

In the present setup, only the rotor is considered, omitting the tower, hub and nacelle, with a total of three overlapping mesh groups, see Figure 4. The omission of these elements is expected have a minor effect, supported by the study (Guma et al., 2021) studying the same rotor represented both as a rotor only and as a full turbine. Near the rotor, an O-O type mesh is grown from the blade surface, extruding ≈15m, discretized  with 128 cells from the surface using the  grid generator HypGrid (Sørensen, 1998). The first cell adjacent to the rotor surface has a height of $1 \cdot 10^{-6}$m to ensure a $y^+$ of less than 1. Each blade is represented through 128 grid points spanwise and 256 chordwise. The blade tip and grid around a blade section are presented in Figure 5. The rotor diameter, D, is ≈80m.

Around the rotor mesh, a  cylindrical disc mesh is constructed with pre-cut holes around the blades. This mesh rotates along with the rotor mesh, speeding up the hole-cutting algorithm, as the holes move along with the rotor. Thereby, the

need of searching for hole, fringe and donor cells between rotor and disc mesh for each time step is avoided, as the relations between the two meshes remain the same.

 All deformation from the rotor

350  is propagated to the rotor mesh in such a way, that only cells that  lie inside the hole region of the overlapping disc mesh deform. This is done to avoid deformation of the donor cells, keeping the interpolation coefficients between fringe and donor cells unaltered. Through this simplification, there is no need for updating communication tables for donor and receiver cells between the rotor and disc mesh  as these also rotate together. This choice, however, necessitates the hole of the disc mesh to be far enough from the surface to leave room for the deformation of mesh cells without impairing the cell quality.

355 In the present setup the holes are 17m wide in the rotor axis direction, where the main deformation is present, located with the undeformed rotor in the center. Displacements are propagated to the volume mesh, such that points within the inner 15% of the grid curve length normal to the surface are moved as solid body motion to ensure no change of quality of the inner cells resolving the high gradient flow. Further out, from 15% to 40% the volume blend factor is linearly decaying from 1 to 0, such that points from 40% grid curve length and out are unchanged to avoid changes to donor cells. Note that communication tables

360 and hole-cutting still need update between disc mesh and background mesh as the latter is static. The disc and rotor grids are similar to the setup used in (Grinderslev et al., 2021), however in this study the background grid has changed to be suitable for LES simulations by using rectangular cells with low cell stretching in the area of focus.

The background domain is a box of 1000 (12.5D) × 600 (7.5D) × 1350 (16.9D) meters (width × height × length) using 352×256×640 cells adding up to ≈58M cells. A concentration of cells is present in the cross flow directions around the rotor

365 area down to 1m side lengths, see Figure 4 (right). Cells in the flow direction are kept constant of ≈1.4m from inlet to the rotor and 6D behind it, before stretching towards the outlet. Boundary conditions are velocity inlet, outlet assuming fully developed flow, and symmetry conditions (slip walls) on sides and top boundaries. The ground has a no-slip wall condition, but with the SG wall model as in the precursor simulation. The rotor is placed ≈4.38D from the inlet, ≈6.25D from sides and top and ≈12.5D from the outlet.

370 A total of 78M cells are used for the combined setup.

[Figure]

**Figure 4.** Grids used for simulations. Left: side view, right: front view. Red cells show receiver cells of overlapping grids. Blue: rotor grid, orange: disc grid, black: background grid. Entire background grid is not shown.

[Figure]

**Figure 5.** Near rotor mesh at 25m span and surface discretization at tip. Only every second line shown

**3 Results**

**3.1 Precursor simulation**

A total of 9750 seconds were simulated for the precursor simulation, of which the final 1000 seconds (equivalent to $\approx$ 270 rotor revolutions) were sampled for statistical post processing, in a period where the developed flow profile suffi-

375   ciently matched the desired profile. The precursor was run in three grid levels with varying time steps. First, the coarse period ($\delta z=\delta y=\delta x \approx$ 16m, $\Delta t$=1.0 sec), a medium period ($\delta z=\delta y=\delta x \approx$ 8m, $\Delta t$=0.5 sec) and finally a fine period ($\delta z=\delta y=\delta x \approx$ 4m, $\Delta t$=0.25 sec). The full precursor simulation was conducted on 1728 AMD EPYC 2.9GHz processors on the computer cluster of DTU, and lasted $\approx$45 wall clock hours. The sampling was conducted in the fine phase only as depicted in

Figure 6 along with the spectra at three different altitudes. As seen, the turbulence is well resolved with a  decent inertial subrange following the Kolmogorov spectrum law with a decaying slope of -5/3.

[Figure]

**Figure 6.** Left: Time series of wind speed, $W$, at three altitudes approximately matching the rotor bottom, center and top altitudes. Right: spectra of wind speed time series (fine resolution period only) using the Welsh estimate

From the sampling plane, depicted in Figure 2, the wind speed profiles of $W$ were extracted and horizontally and temporally averaged $\pm$ 1D from the rotor position in the cross plane direction as depicted in Figure 7 (left). As seen, the relative difference of the averaged profile and the DanAero log-law fit match well with a maximum of 8% at $\approx$ 14m, which corresponds to only $\approx$0.5m/s at the specific altitude. One difference to note, however, is the larger standard deviation, and thereby turbulence intensity, of the sampled flow, with fluctuations that supersede the DanAero measurements. The complexity of fitting both mean profile and turbulence intensity between measurements and LES simulations is high. In this specific case, the assumption of neutral stratification in the simulation, while no knowledge about stratification being available from the measurements, likely plays a role in the capabilities to match results. This was the best match obtained after multiple calibration attempts, considering both mean profile and turbulence intensity.

Figure 8 depicts the resulting resolved and SGS flow shear stresses and resolved friction velocity, $u^*$.

[Figure]

**Figure 7.** Left: Horizontal and temporal average profile $\mu_W \pm 1 \sigma$ (solid red and red patch), DanAero measurements and fitted log-law (blue errorbars and dashed). Horizontal averaging based on flow from $\pm 1D$ from the rotor center on the sampled flow plane. Right: Relative error between log-law fit and $\mu_W$ profile

[Figure]

**Figure 8.** Left:Horizontal averages of resolved and SGS shear stresses at 9250 sec. Right: Resulting resolved friction velocity $(u*)^2 = \sqrt{vw^2 + uv^2}$

**3.2 Successor simulation**

In the following, the results of the successor simulations are presented. First, the new turbulence model is compared to the same setup using only the IDDES turbulence model assuming a elastically stiff configuration. Further, results from simulations using the hybrid model with and without flexibility of the blades are presented to study the effect of the blade elasticity. For the initial phase 1 (see Figure 3), simulations were conducted on 1189 AMD EPYC 2.9GHz processors, while coupled simulations

395

(phases 3 and 4 in Figure 3) were conducted on 793 Intel Xeon 2.8GHz processors. The initialization simulations took in the order of ≈26 wall clock hours, while the coupled simulations lasted for ≈180 wall clock hours per simulation.

[Figure]

**Figure 9.** Isobars of Q-criterion=0.4 colored with value of flow velocity $W$

Figure 9, shows the Q-criterion=0.4 (Hunt et al., 1988) of the flow, visualizing the turbulent structures up- and especially downwind of the rotor. As seen, the tip vortices in the wake are quickly broken up into smaller structures by the surrounding

400    turbulent flow. This is also visible in Figure 10 showing the  flow velocity $W$ at multiple downstream positions.

[Figure]

**Figure 10.** Velocity  *W* downstream of rotor.

**3.2.1 Impact of turbulence model**

To study the impact of the presented turbulence model on the flow, simulations with the hybrid LES/IDDES blending enabled along with pure IDDES simulations are conducted. In the pure IDDES simulation, a slip wall condition is used on the terrain
405 surface, contrary to the log-law used for the LES/IDDES hybrid model. Simulations with and without the rotor present were simulated. In the empty setup, the hybrid model acts as a pure Deardorff LES model, as no blending region is defined. For all simulations, inflow is interpolated from the LES precursor planes to ensure identical inlet conditions. In the simulations comparing turbulence models, only the CFD code has been used, meaning that no flexibility of the blades is considered.

Firstly, the empty setups are presented in Figure 11 showing the velocity component $W$ at a vertical plane
410 aligned with the flow direction intersecting the rotor center, for the simulations with Deardorff and IDDES turbulence modelling at the same time instance. From the planes, instantaneous velocity profiles and turbulence intensity (TI) profiles are extracted along the dashed lines, which are shown in Figure 12. Both simulations show very comparable results. As seen, the velocity and TI profiles, extracted 96m from the inlet, are practically identical, while a  discrepancy is seen further downstream in the domain as a result of changing the turbulence and wall models. While both
415 turbulence models perform close to identically for the present single turbine study, it could be speculated that this would not be the case when considering larger domains, for instance when studying multiple turbines at once. In that case, the differences

in turbulence and wall modelling will likely result in different flow fields, due to the longer distances covered. Considering temperature effects on the flow might likewise reveal differences between the turbulence models. This, in terms of temperature flux differences, along with the Deardorff model considering the stability in the mixing length scale, which is not the case in 420 the IDDES model.

[Figure]

**Figure 11.** Instantaneous sections of flow velocity $W$  for hybrid (left) and IDDES (right) simulations. Black dashed lines indicate the locations of the profiles presented in Figure 12

[Figure]

**Figure 12.** Instantaneous  sampled profiles at time=129.6 seconds. Top row: flow velocity $W$, bottom: turbulence intensity (TI). Samples extracted from planes close to the inlet (96m from inlet) and far downstream (691m from inlet)  see dashed lines of Figure 11.

[revised manuscript text omitted]

**5 Future studies**

As it was shown in the present study, the developed hybrid turbulence model resulted in practically identical loading of the rotor as the IDDES model alone. Relevant future studies would be to investigate when this is not the case. This could for instance be simulations including stable/unstable stratification and/or Coriolis force. Here, the IDDES model will probably be insufficient to capture the effects, as the model is calibrated for aerodynamics mainly and not ABL flows. The Deardorff LES model, however, is calibrated for such flows and the mixing length scale depends on the stratification, as it is reduced for stable cases. The two models also model temperature effects differently as the flux, $\tau_{\theta,w}$, is based on different weights of the turbulent viscosity. 
[revised manuscript text omitted]